# The lncRNA *SNHG26* drives the inflammatory-to-proliferative state transition of keratinocyte progenitor cells during wound healing

Dongqing Li [1] ✉, Zhuang Liu[2,6], Letian Zhang[2,6], Xiaowei Bian [2,6], Jianmin Wu [3], Li Li[1], Yongjian Chen[2], Lihua Luo [2], Ling Pan[1], Lingzhuo Kong[1], Yunting Xiao[1], Jiating Wang[1], Xiya Zhang[1], Wang Wang[4], Maria Toma [2], Minna Piipponen[2], Pehr Sommar[5] & Ning Xu Landén [2] ✉

The cell transition from an inflammatory phase to a subsequent proliferative phase is crucial for wound healing, yet the driving mechanism remains unclear. By profiling lncRNA expression changes during human skin wound healing and screening lncRNA functions, we identify *SNHG26* as a pivotal regulator in keratinocyte progenitors underpinning this phase transition. *Snhg26*-deficient mice exhibit impaired wound repair characterized by delayed re-epithelization accompanied by exacerbated inflammation. Single-cell transcriptome analysis combined with gain-of-function and loss-of-function of *SNHG26* in vitro and ex vivo reveals its specific role in facilitating inflammatory-to-proliferative state transition of keratinocyte progenitors. A mechanistic study unravels that *SNHG26* interacts with and relocates the transcription factor ILF2 from inflammatory genomic loci, such as *JUN, IL6, IL8*, and *CCL20*, to the genomic locus of *LAMB3*. Collectively, our findings suggest that lncRNAs play cardinal roles in expediting tissue repair and regeneration and may constitute an invaluable reservoir of therapeutic targets in reparative medicine.

Tissue repair requires three sequential and overlapping phases, i.e., inflammation, proliferation, and remodeling[1,2]. The transition from the inflammatory to the proliferative phase represents a key step during wound healing, as failure of this transition leads to impaired re-epithelialization accompanied by persistent inflammation, which has been recognized as a pathological hallmark of chronic wounds[1,3–5]. Therefore, gaining a deeper understanding of the mechanisms that

drive this inflammation-proliferation phase transition may lead to innovative approaches to expedite tissue repair.

Keratinocytes constitute the major cell type of the epidermis and play important roles in both the inflammatory and proliferative phases of wound healing. Upon skin injury, keratinocytes are activated and release a diverse range of inflammatory mediators that are crucial for the recruitment and activation of immune cells to the wound sites[6–8]. In the

[1]Key Laboratory of Basic and Translational Research on Immune-Mediated Skin Diseases, Chinese Academy of Medical Sciences; Jiangsu Key Laboratory of Molecular Biology for Skin Diseases and STIs; Institute of Dermatology, Chinese Academy of Medical Sciences and Peking Union Medical College, 210042 Nanjing, China. [2]Dermatology and Venereology Division, Department of Medicine Solna, Center for Molecular Medicine, Karolinska Institutet, 17176 Stockholm, Sweden. [3]Key Laboratory of Laboratory Medicine, Ministry of Education, Institute of Genomic Medicine, Wenzhou Medical University, 325035 Wenzhou, China. [4]Shanghai Key Laboratory of Regulatory Biology, School of Life Sciences, East China Normal University, 200241 Shanghai, China. [5]Department of Plastic and Reconstructive Surgery, Karolinska University Hospital, 17176 Stockholm, Sweden. [6]These authors contributed equally: Zhuang Liu, Letian Zhang, Xiaowei Bian. ✉e-mail: dongqing.li@pumcderm.cams.cn; ning.xu@ki.se

following proliferative phase, the immune response of keratinocytes diminishes, and keratinocytes, especially stem/progenitor cells, play central roles in wound re-epithelialization via proliferation and migration[9]. The cell function switch from inflammation to proliferation or migration requires reprogrammed gene expression; however, the molecular machinery steering this phase transition is not well understood.

Long noncoding RNAs (lncRNAs) are expanding the horizon in understanding the regulatory logic encoded in our genome[10]. Depending on their subcellular localization and specific interactions with DNA, RNA, and proteins, lncRNAs have been shown to modulate chromatin structure and function as well as the transcription of neighboring and distant genes and to affect RNA splicing, stability, and translation[11]. Notably, several lncRNAs have been shown to regulate keratinocyte progenitor proliferation (e.g., *PRANCR*[12]) and differentiation (e.g., *TINCR* and *ANCR*[13,14]). We aimed to understand the roles of lncRNAs in the inflammatory-to-proliferative phase transition and identify lncRNA(s) crucial in wound repair.

In this study, we tracked gene expression in human skin wound healing and identified 20 evolutionarily conserved lncRNAs with significant changes during the inflammatory and proliferative phases. Among these, small nucleolar RNA host gene 26 (*SNHG26*), a lncRNA derived from snoRNA host genes (SNHGs), emerged as a key regulator facilitating the transition from inflammation to proliferation in keratinocyte progenitors. Mechanistically, *SNHG26* interacts with ILF2, shifting this transcription factor (TF) from inflammatory genomic loci to the *LAMB3* locus, thereby facilitating the transcriptional program switch from inflammation to re-epithelialization.

## Results

### LncRNA expression profiling and functional screening in human skin wound healing

To probe gene expression during human skin wound healing, we created full-thickness excisional wounds on the skin of healthy volunteers ($n = 26$) and then collected wound-edge tissues from the same donors at each healing stage, i.e., inflammation (day 1 wounds, NW1), proliferation (day 7 wounds, NW7), and remodeling (day 30 wounds, NW30) (Fig. 1a and Supplementary Data 1). We conducted whole-transcriptome analysis using polyadenylation-independent total RNA sequencing (RNA-seq) on matched skin, NW1, and NW7 samples from five donors, identifying 342 lncRNAs with significantly altered expression during wound repair (one-way ANOVA, $p \leq 0.005$) (Fig. 1a and Supplementary Data 2). We focused on 20 lncRNAs conserved in humans and rodents[15,16], as evolutionary conservation suggests crucial functional roles[17]. After transfection of Lincode siRNAs targeting each of these lncRNAs, we assessed their impact on the inflammatory response, proliferation, and migration of human keratinocyte progenitors (Fig. 1b). Silencing *SNHG26*, *RMRP*, *MALAT1*, *CRNDE*, or *SNHG6* increased TNFα-induced production of the inflammatory cytokines IL6 and CCL20. Moreover, downregulating *SNHG26*, *SNHG6*, or *LOC101929709* decreased cell proliferation, while silencing *SNHG26*, *WDFY3-AS2*, *CRNDE*, *SOX9-AS1*, or *GAS5* reduced keratinocyte migration. Notably, among these 20 conserved lncRNAs, knockdown of SNHG26 exhibited the most pronounced proinflammatory, antiproliferative, and antimigratory effects on keratinocytes.

### Characterization of SNHG26 expression in skin wound healing

*SNHG26* is an intergenic lncRNA encoded on human chromosome 7 (GRCh37/hg19, chr7: 22,893,745-22,901,055) and mouse chromosome 5 (GRCm38/mm10, chr5:23,850,597-23,855,038, annotated as *2700038G22Rik*) (Fig. 1c, Supplementary Fig. 1a and 1b). Comparison of the RNA sequence of human and mouse SNHG26 showed poor sequence similarity (Supplementary Fig. 2). However, both the human and mouse *SNHG26* genes are located between the *TOMM7* and *FAM126A* coding genes, suggesting interspecies syntenic conservation (Fig. 1c, Supplementary Fig. 1a and 1b).

To investigate the expression of SNHG26 during human and mouse wound healing, we collected samples from the wound edges at various stages of healing including inflammation, proliferation, and remodeling phase. QRT-PCR analysis revealed that inflammation (S100A7 and S100A8), proliferation (MKI67), and remodeling (COL1A1 and COL3A1), which expression levels peak at the NW1, NW7, and NW30, respectively, supporting the associations between selected time points and phases of human wound healing (Supplementary Fig. 3a). Similarly, in the mouse wound model, we showed peaked expression of inflammatory genes (Il6 and Il1a), proliferative gene (Mki67), and remodeling process related gene (Eln) in NW3, NW7, and NW10 wounds, respectively, linking these time points with corresponding healing stage (Supplementary Fig. 3b–e). In both humans and mice, *SNHG26* expression temporarily increased during the inflammatory and proliferative phases of wound healing, returning to baseline levels during remodeling phase (Fig. 1d, e). To identify the primary cell type(s) expressing *SNHG26* in human skin and wounds, we isolated epidermal CD45− cells (primarily composed of keratinocytes) and CD45+ cells (leukocytes), dermal CD90+ cells (fibroblasts), CD14+ cells (macrophages), and CD3+ cells (T cells) from matched skin and day 7 wounds of five healthy donors (Supplementary Data 1). *SNHG26* was found in various skin cell types, but keratinocytes showed the most significant upregulation during wound healing (Fig. 1f). This observation was further validated by spatial transcriptomics (Fig. 1g, Supplementary Fig. 3f, g) and fluorescence in situ hybridization analysis (FISH, Fig. 1h) of human acute wounds, indicating that the injury-induced transient upregulation of *SNHG26* expression predominantly occurred in wound-edge basal keratinocytes. Notably, the *SNHG26* gene locus also contains a small nucleolar RNA *SNORD93* in both humans and mice (Fig. 1c, Supplementary Fig. 1a and b), but only *SNHG26* showed increased expression in keratinocytes during wound repair (Supplementary Fig. 3h and i). Moreover, *SNHG26* expression was reduced in human diabetic foot ulcers and diabetic mouse wounds compared to acute wounds (Supplementary Fig. 3j–l), highlighting its relevance in wound healing and chronic wound pathology.

We further characterized the molecular features of *SNHG26* in keratinocyte progenitors. The Cap Analysis of Gene Expression (CAGE) project[18] previously located the transcription start site (TSS) of *SNHG26* at GRCh37/hg19 chr7:22893816-22893881 (Fig. 1c). Our 3′ rapid amplification of complementary DNA ends (RACE) analysis indicated that the SNHG26 sequence matched the transcript ENST00000415611.9 from Ensembl, with a length of 3100 nt (Fig. 1c). This finding was further validated through Northern blot analysis (Fig. 1i). Furthermore, FISH and nucleus-cytoplasm fractionation assay revealed the nuclear localization of *SNHG26* in both cultured keratinocyte progenitors and basal keratinocytes in human skin (Fig. 1h, j, k). Moreover, by separating the poly(A)$^+$ fraction from the poly(A)$^-$ RNA fraction, we found *SNHG26* to be a polyadenylated RNA (Supplementary Fig. 3m). With the Coding Potential Calculator 2 algorithm[19], we confirmed the non-coding potential of *SNHG26* (Supplementary Fig. 3n). Additionally, we determined that the half-life of *SNHG26* in keratinocyte progenitors was 2.2 hours (Supplementary Fig. 3o). In summary, *SNHG26* is a conserved, polyadenylated, nuclear lncRNA transiently upregulated in wound-edge basal keratinocytes.

### Impaired wound healing in Snhg26-deficient mice

Considering the potential role of *SNHG26* in regulating keratinocyte inflammation, proliferation, and migration (Fig. 1b), we proceeded to investigate whether *SNHG26* upregulation is indispensable for the wound healing process. To examine this, we generated Snhg26-knockout (KO) mice by deleting the 8 kb genomic locus (Supplementary Fig. 4a, b). These KO mice exhibited a generally normal phenotype with a slightly thinner epidermis (Supplementary Fig. 4c). However, upon skin injury, we observed a significant 40% delay ($p = 0.0001$) in wound healing in the *Snhg26*-KO mice compared to control mice

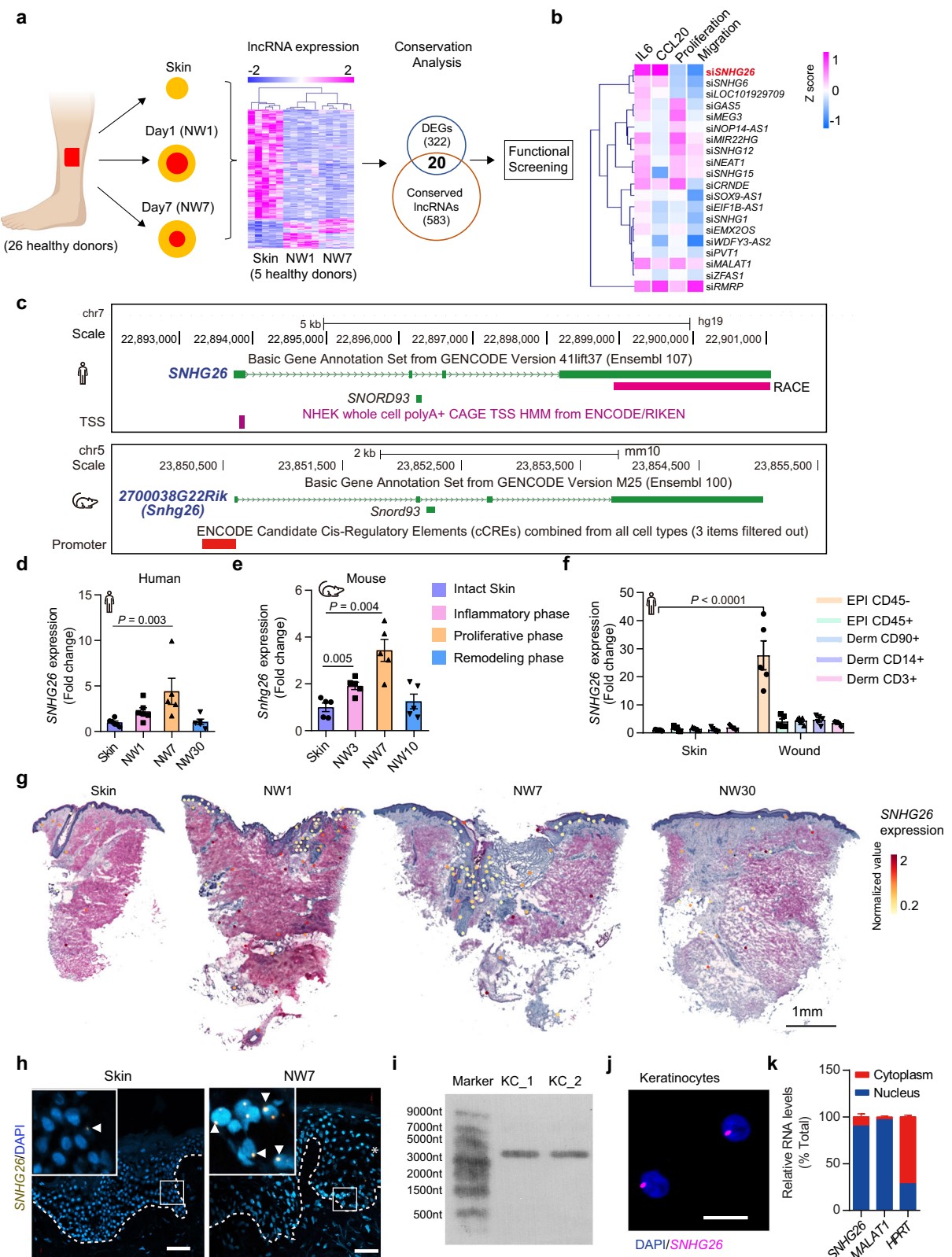

(Fig. 2a). Recognizing that compensatory mechanisms might be at play in this model with constitutive *Snhg26* deletion, we also intradermally injected *Snhg26* antisense oligos (*Snhg26*-ASOs) at wound edges in wild-type (WT) mice (Supplementary Fig. 4d–f). These ASOs were designed to block the injury-induced upregulation of *Snhg26* by facilitating RNaseH1-dependent cleavage of *Snhg26* transcripts[20]. Notably, these ASOs effectively reduced *SNHG26* expression in the epidermal compartment, while the dermal compartment remained unchanged (Supplementary Fig. 4f). Similar to the *Snhg26*-KO mice, mice treated with *Snhg26*-ASOs exhibited a 37.5% delay (*p* = 0.003) in wound healing compared to those receiving scrambled ASOs, emphasizing the crucial role of *Snhg26* transcripts in wound repair (Fig. 2b).

To uncover the gene expression changes linked to reduced healing in *Snhg26*-KO mice, we compared epidermal gene expression

**Fig. 1 | LncRNA expression profiling and functional screening in human skin wound healing. a** Profiling human skin lncRNA expression changes during wound healing by RNA-seq. **b** Functional screening of 20 differentially expressed and conserved lncRNAs in keratinocyte progenitors for cell proliferation, migration, and TNFα-induced inflammatory cytokine expression. **c** Human and mouse SNHG26 genomic loci. **d**, **e** qRT–PCR analysis of *SNHG26* expression in human (*n* = 6 in Skin and NW1, *n* = 5 in NW7 and NW30) (**d**) and mouse acute wounds (*n* = 5) (**e**). **f** qRT-PCR analysis of *SNHG26* in epidermal (EPI) CD45− and CD45+ cells and in dermal (Derm) CD90 + , CD14 + , and CD3+ cells, which were isolated from the skin and day-7 wounds of healthy donors (*n* = 5). **g** Visualization of *SNHG26* expression in human acute wounds analyzed by spatial transcriptomics. **h** In situ hybridization

analysis of *SNHG26* in human skin and day-7 acute wounds. The asterisk (*) indicates the wound edge. Nuclei were stained with DAPI. Scale bar, 25 μm. **i**, **j** Northern blot analysis **i** and In situ hybridization analysis **j** of *SNHG26* in cultured human keratinocyte progenitors (KC). Scale bar, 10 μm. **k** qRT-PCR analysis of *SNHG26*, *MALAT1*, and *HPRT* in nuclear (blue) and cytosolic (red) compartments of human keratinocyte progenitors (*n* = 3 biological replicates). Data are shown as mean ± SD (**d**–**f**, **k**). The data were analyzed by one-way ANOVA analysis (**d**, **e**) or unpaired two-tailed Student's t test (**f**). Panel a was created with BioRender.com and released under a Creative Commons Attribution-NonCommercial-NoDerivs 4.0 International license (https://creativecommons.org/licenses/by-nc-nd/4.0/deed.en).

between KO (*n* = 3) and WT mice (*n* = 3). Microarray analysis identified 253 and 431 differentially expressed genes (DEGs; fold-change ≥2 or ≤ -2, *p* value < 0.05) in the *Snhg26*-KO mice skin and day 3 wound edge, respectively (Supplementary Fig. 4g). Gene Ontology (GO) analysis revealed that the downregulated genes in *Snhg26*-KO skin epidermis were mainly associated with epidermal differentiation, while no enriched GO term was found among the upregulated DEGs (Supplementary Fig. 4h). In the wound-edge epidermis, *Snhg26*-KO mice exhibited upregulated inflammation-related genes and downregulated genes linked to skin development and cell migration (Fig. 2c). Gene set enrichment analysis (GSEA) showed that upregulated genes were associated with leukocyte migration, while downregulated genes were linked to epithelial cell proliferation and migration in the wound-edge epidermis of *Snhg26*-KO mice (Fig. 2d).

We also noted reduced expression of *Krt16* and *Sprr1b* in the wound-edge epidermis of *Snhg26*-KO mice compared to controls, as confirmed by qRT–PCR analysis of additional skin and wound epidermal samples (Fig. 2e). Both Krt16 and Sprr1b are known for their roles in keratinocyte proliferation and migration[21,22]. The decreased expression of these genes could potentially contribute to delayed re-epithelialization, as supported by histomorphometry analysis of newly formed epithelial tongues (Fig. 2f). Furthermore, we observed significant upregulation of proinflammatory cytokines, including *Il6* and *Il1b*, in the wound-edge epidermis of *Snhg26*-KO mice (Fig. 2g) and increased macrophages (CD68 + ) in the dermal wound bed of *Snhg26*-KO mice (Fig. 2h). Similarly, mice treated with *Snhg26*-ASOs exhibited elevated levels of inflammatory cytokines (*Il6*, *Ccl20*, *Ccl2*, and *Ccl5*) and increased macrophages (CD68 + ) in the wounds (Supplementary Fig. 5).

In summary, our study, utilizing *Snhg26*-KO mice and mice with transient *Snhg26* inhibition at wound edges, provides conclusive evidence of *Snhg26*'s in vivo role in wound repair. Its deficiency not only delayed re-epithelialization but also exacerbated inflammatory response.

## Single-cell transcriptome analysis of delayed wound repair in Snhg26-KO mice

To gain deeper insights into the delayed wound repair in *Snhg26*-KO mice, we conducted single-cell RNA sequencing (scRNA-seq) analysis on skin and day 3-wound tissues from *Snhg26*-KO mice (*n* = 3) and control littermates (*n* = 3). Unsupervised clustering identified 23 cell clusters, including keratinocytes (basal stem and progenitors: C1-C4; spinous: C5; granular: C6; hair follicle keratinocytes: C7-C10), melanocytes (C11), fibroblasts (C12-C14), sebaceous gland cells (C15), vascular endothelial cells (C16-C17), muscle cells (C18-C19), macrophages (C20), T helper cells (C21), γδ T cells (C22), and Langerhans cells (C23) (Fig. 3a, Supplementary Fig. 6a). Each of these cell clusters contained cells from all the investigated samples (Supplementary Fig. 6b). Notably, our analysis revealed that *Snhg26* was predominantly expressed in keratinocytes (Fig. 3b, c). While the cellular composition remained similar between the KO and WT mouse skin samples, notable differences were observed in the basal keratinocytes and macrophages in the wounds of *Snhg26*-KO mice compared

to WT mice (Fig. 3d). We also calculated the number of DEGs in each cell cluster between *Snhg26*-KO and WT mice and found that keratinocytes showed the highest number of DEGs between the two groups (Fig. 3e), suggesting the critical role of *SNHG26* in keratinocytes during wound healing.

Consistent with a recent study on mouse wounds using scRNA-seq[23], our results identified four basal stem and progenitor cell clusters: C1 expressed *Col17a1*, a marker enriched in epidermal stem cells; C2 consisted of proliferative basal cells expressing mitosis-related genes, such as *Pcna* and *Mki67*; C3 showed enrichment in glycolysis-related genes, important for keratinocyte migration[24], and specifically expressed the keratinocyte activation markers *Krt6b* and *Krt16*[25]; and C4 expressed genes promoting cell cycle arrest (e.g., *Ovol1*) and inflammation (e.g., the TFs *Fosl1* and *Rel*)[26,27] (Fig. 3f, Supplementary Fig. 7a, b, Supplementary Data 3). We further performed immunostaining for these four different basal cell populations in the wound edge: KRT15 (Bas1), MKI67 (Bas2), GJB2 (Bas3), and KLF5 (Bas4). All four cell populations were successfully detected in the wound edge epidermis (Supplementary Fig. 7c). Functional characteristics of these keratinocyte progenitor clusters, including the proliferative C2, migratory C3, and inflammatory C4 progenitors, were more evident when we compared overall gene expression scores for these processes (Fig. 3g, Supplementary Data 4). Analyzing the functional scores of individual keratinocytes, we observed fewer proliferative or migratory keratinocytes but a higher number of inflammatory keratinocytes in the wounds of *Snhg26*-KO mice compared to WT mice (Fig. 3h–j). This observation was supported by GO analysis of the DEGs in keratinocyte progenitors between the *Snhg26*-KO and WT mice (Supplementary Fig. 8a–c). Furthermore, a cell-cell crosstalk analysis[28] highlighted enhanced CCL and CXCL chemokine signaling from basal keratinocytes to macrophages in *Snhg26*-KO wounds (Fig. 3k, Supplementary Fig. 8d). These findings align with the more numbers of macrophages detected in these wounds by scRNA-seq (Fig. 3d) and IF staining (Fig. 2h). Moreover, we found that the expression of *Krt16* and *Sprr1b* was significantly decreased, while the expression of *Il6* and *Il1b* was increased in the keratinocytes of KO mice wounds compared with WT mice wounds (Supplementary Fig. 8e–h), which is consistence with the results as we detected by qRT-PCR (Fig. 2e and g).

Single-cell analysis allowed us to precisely discern the impact of *Snhg26*-KO in various skin cell types. In dermal fibroblasts, which play critical roles in skin wound healing, especially in modulating the inflammatory response[29], we observed 198 DEGs (Fig. 3e). The downregulated genes in the *Shng26*-KO fibroblasts were associated with the TGF-β regulation of extracellular matrix, while the upregulated genes were related to TNF-α signaling via NF-kB pathway (Supplementary Fig. 8i). Cell-cell crosstalk analysis further confirmed the role of fibroblasts in wound inflammation. Specifically, fibroblasts (Fb1 and Fb2) were identified as the primary cell types sending CCL and CXCL signals in mouse wounds (Fig. 3k and Supplementary Fig. 8d). In *Snhg26*-KO wounds, fibroblasts exhibited heightened CCL signals directed towards keratinocytes (Fig. 3k). Additionally, there was a decreased laminin signal from keratinocytes to fibroblasts in the *Snhg26*-KO mice (Supplementary Fig. 8j).

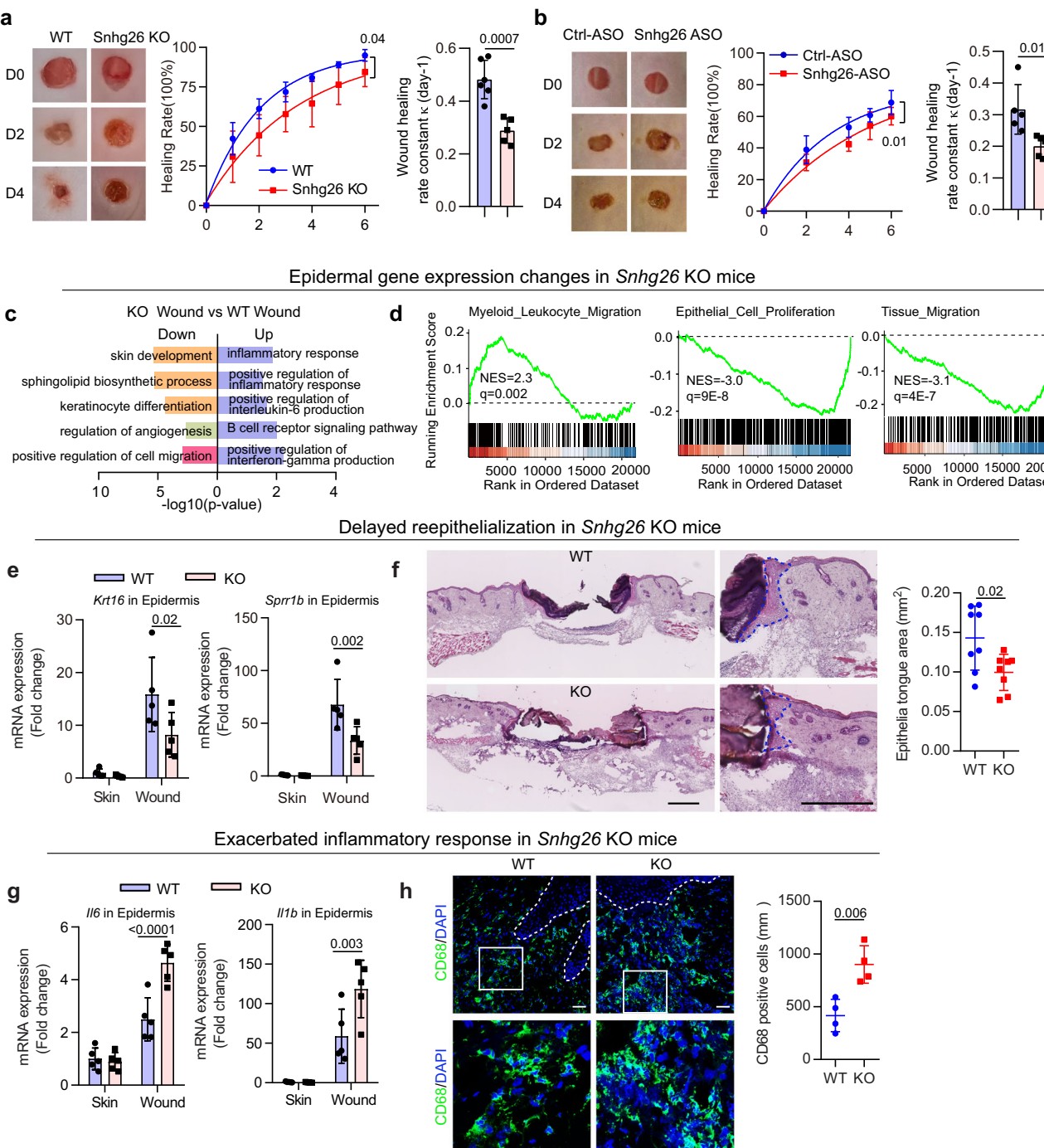

**Fig. 2 | Impaired wound healing in *Snhg26*-deficient mice. a, b** Representative wound images and wound closure rates of *Snhg26*-KO mice (*n* = 5) vs. WT mice (*n* = 6) (**a**) and C57BL/6 mice treated with *Snhg26*-ASOs or control ASOs (*n* = 5/group) (**b**). Wound healing rate constant κ (day−1) was calculated using a one-phase decay model in GraphPad. **c** Gene Ontology analysis of differentially expressed genes in the wound-edge epidermis of *Snhg26*-KO mice vs. WT mice. **d** Gene set enrichment analysis of wound-edge epidermal gene expression changes in *Snhg26*-KO mice vs. WT mice. Shown are the false discovery rate (FDR) value and normalized enrichment score (NES). **e** qRT–PCR analysis of *Krt16* and *Sprr1b* expression in the skin and wound-edge epidermis of *Snhg26*-KO and WT mice (*n* = 5). **f** Hematoxylin and eosin staining of *Snhg26*-KO and WT mouse wounds. Areas of the wound tongue epithelium (marked with dotted lines) were quantified (*n* = 8). Scale bar, 200 μm. **g** qRT–PCR analysis of *Il6* and *Il1b* expression in mouse skin and wound-edge epidermis (*n* = 5). **h** Immunofluorescence analysis of CD68+ cells in mouse wounds (*n* = 4). The white dashed line demarcates the epidermis and dermis border. Scale bar, 20 μm. Data are shown as mean ± SD from two to three independent experiments (**a**, **b**, **e**–**h**). The data were analyzed by two-way ANOVA (**a**, **b**, **e** and **g**), two-sided Fisher's exact test (**c**) or two-tailed Student's t test (**f** and **h**).

In summary, *Snhg26* disruption skewed keratinocyte progenitors towards an inflammatory state and away from proliferation or migration. While *Snhg26* had a lesser impact on dermal fibroblasts, the changes in keratinocyte-fibroblast communication collectively contributed to delayed wound healing.

## SNHG26 inhibits the human keratinocyte inflammatory response and promotes re-epithelialization

Next, we explored the physiological relevance of *SNHG26* in human keratinocytes and skin wound healing. We silenced *SNHG26* expression in human keratinocyte progenitors by transfecting *SNHG26*-ASOs

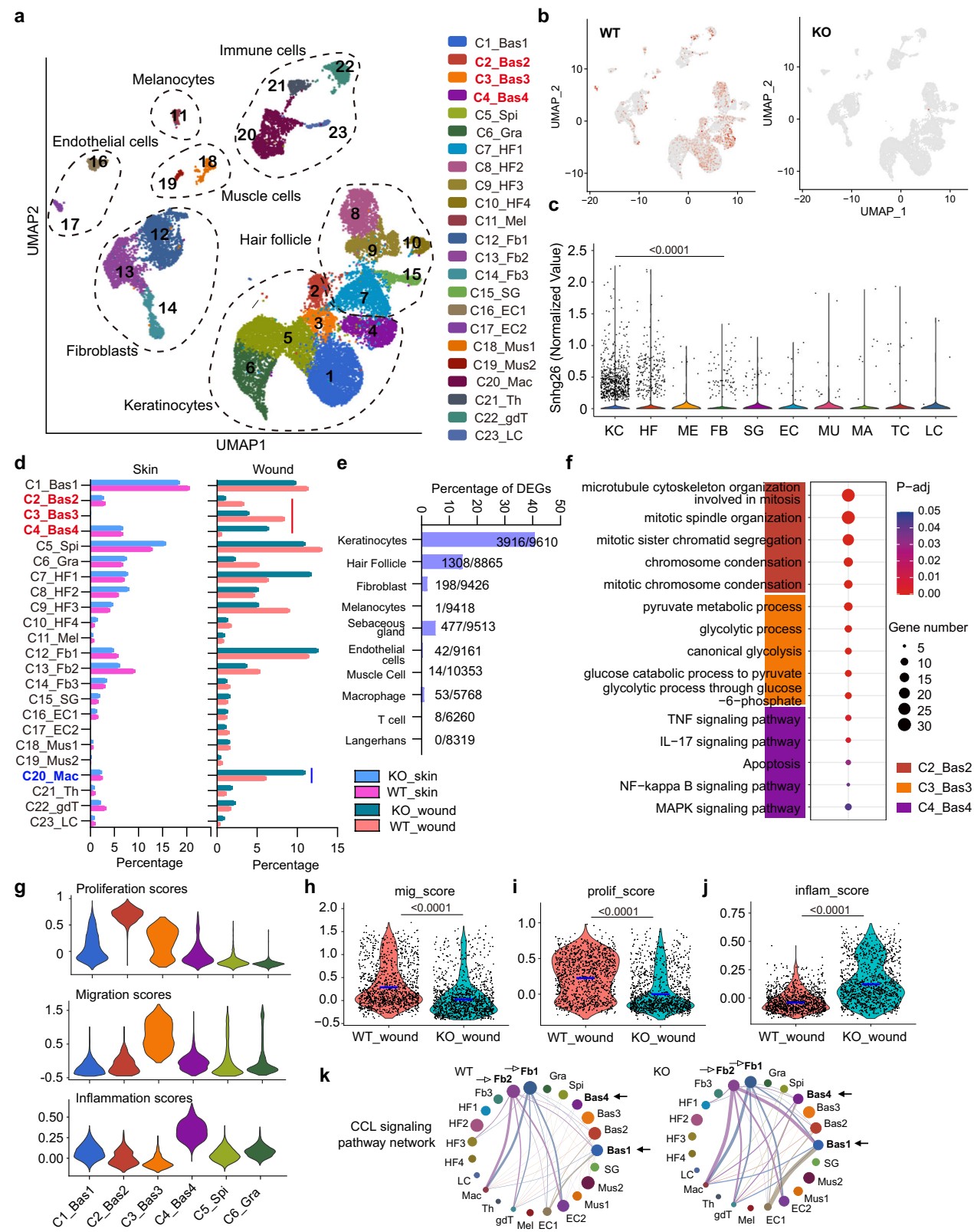

followed by TNFα treatment to trigger an inflammatory response[30]. This ASO was designed to target the exons of *SNHG26*, and it led to a significant reduction in *SNHG26* expression while leaving *SNORD93* unaffected (Supplementary Fig. 9a, b). Microarray analysis of these cells revealed gene expression changes (ANOVA, *P* < 0.001) grouped into five patterns (Fig. 4a). Module 2 (M2) contained 898 upregulated genes related to inflammation. M3 comprised 123 TNFα-induced genes

that were further upregulated by *SNHG26* knockdown. M4 and M5 included 1157 and 918 downregulated genes related to mitosis.

We confirmed expression changes in genes associated with inflammation, including *IL6*, *IL8*, and *CCL20*, which are induced in response to skin injury and play crucial roles in the inflammatory process[31,32]. Silencing *SNHG26* increased their expression under both basal and TNF-α-triggered inflammatory conditions (Fig. 4b).

**Fig. 3 | Single-cell transcriptomic analysis of impaired wound repair in *Snhg26*-KO mice. a** UMAP plot and cluster annotation of 21491 cells from skin and wounds of *Snhg26*-KO and WT mice (*n* = 3/group). **b** UMAP plot showing the expression of *Snhg26* in *Snhg26*-KO and WT mice. **c** Violin plot showing the expression of *Snhg26* in different cell types in WT mice. **d** Comparison of skin and wound cell composition between the KO and WT mice. Red line indicates keratinocyte clusters with changed cell composition in the wound of KO mice. Blue line indicates immune cell clusters. **e** The proportion of DEGs [log2(FC) ≥ 0.3, -log10(FDR) > 2] within all expressed genes in individual cell clusters in wounds of *Snhg26*-KO and WT mice.

**f** GO and KEGG analysis of the top 200 marker genes in basal keratinocyte cell clusters. **g** The proliferation, migration, and inflammatory scores were plotted for each keratinocyte cluster. **h–j** Violin plots showing keratinocyte migration (**h**), proliferation (**i**) and inflammation scores (**j**) in Snhg26-KO and WT mice wound basal keratinocytes. **k** Circle plots displaying the CCL signaling network in *Snhg26*-KO and WT mice wounds. The changed signals in the KO mice compared to the WT mice were highlighted with arrows. The data were analyzed by two-sided post hoc test (**c**), two-sided Fisher's exact test (**f**) or two-sided Mann-Whitney U test (**h–j**).

Conversely, nuclear overexpression of *SNHG26* using a pZW1-snoVector reduced TNF-α-induced *IL6*, *IL8*, and *CCL20* expression (Fig. 4c, Supplementary Fig. 9c)[33]. However, cytoplasmic overexpression of *SNHG26* using a pcDNA3.1 vector had no discernible effect on these inflammatory genes (Supplementary Fig. 9d–g), highlighting the importance of subcellular localization in SNHG26's function.

Furthermore, we substantiated our findings by confirming that the silencing of *SNHG26* reduced, whereas overexpression of *SNHG26* increased the migration and proliferation of human keratinocyte progenitors. This was evident in scratch wound assays (Fig. 4d, f), cell proliferation assays (Fig. 4e), and colony formation assays (Fig. 4g, Supplementary Fig. 9h). Next, we extended our analysis to human ex vivo wound closure, which is an in vivo-like and clinically relevant model for analyzing human wound re-epithelization[34,35]. We topically applied *SNHG26*-ASOs (Fig. 4h, i, Supplementary Fig. 9i, j) or its overexpression plasmid sno-SNHG26 (Fig. 4j, k) mixed in a lipid-based transfection reagent to the wounds. This intervention specifically modulated epidermal SNHG26 levels, leaving dermal levels unaffected (Supplementary Fig. 9k). We found that *SNHG26*-ASOs reduced wound re-epithelialization, while overexpression of *SNHG26* enhanced this process (Fig. 4h–k).

In addition, we conducted experiments to distinguish the roles of *SNHG26* and *SNORD93*, both sharing the same genomic locus (Fig. 1c). Using CRISPR-Cas9 technology, we generated a *SNHG26* knockout in hTERT-immortalized human keratinocyte cell line (Ker-CT) (Supplementary Fig. 9l, m). Subsequently, we reinstated the expression of either *SNORD93* or *SNHG26* through the transfection of their respective overexpressing plasmids (Supplementary Fig. 9n, o). Notably, *SNORD93* had no impact on inflammatory responses or migration, whereas *SNHG26* reintroduction exhibited anti-inflammatory and pro-migratory effects, confirming the specific role of *SNHG26* (Supplementary Fig. 9p-s).

Collectively, our mouse and human data jointly demonstrated the evolutionary conservation of *SNHG26* in both expression and function: it is an injury-induced lncRNA in keratinocyte progenitors and plays pivotal roles in inhibiting cell inflammatory response while promoting cell proliferation and migration and wound re-epithelization.

## Silencing SNHG26 increase binding of ILF2 to inflammatory genomic loci

Long non-coding RNAs (lncRNAs) regulate gene expression through both cis and trans mechanisms. To study if SNHG26 regulate the genes near its genomic locus, we investigated the expression of SNHG26-neighboring genes, *Tomm7* and *Fam126a*, in both WT and SNHG26-KO mice under hemostatic and wounding conditions. We observed no significant changes in the expression levels of these genes between the WT and SNHG26-KO mice (Supplementary Fig. 10a, b). Similarly, knockdown of SNHG26 in primary keratinocytes did not affect the expression of *TOMM7* and *FAM126A* (Supplementary Fig. 10c, d). Thus, SNHG26 may exert it regulatory function by in trans.

To elucidate *SNHG26*'s molecular mechanism, we performed RNA pull-down assay in human keratinocyte progenitors (Fig. 5a). An SDS–PAGE analysis revealed a protein band at ~45 kDa specifically copurified with *SNHG26* (Fig. 5b). The mass spectrometry (MS) analysis

identified multiple proteins (Supplementary Data 5). We excluded keratins from this list, as they are commonly regarded as contaminants in MS studies[36]. We were particularly interested in proteins localized to the nucleus, given that SNHG26 is primarily expressed in this compartment. Therefore, we focused on interleukin enhance-binding factor 2 (ILF2), which is a transcription factor locating in the nucleus. We further confirmed the interaction between SNHG26 and ILF2 with another set of independent samples by Western blotting (Fig. 5c). The reciprocal binding of *SNHG26* and ILF2 protein was validated via RNA immunoprecipitation (RIP) using the anti-ILF2 antibody, which showed that *SNHG26*, but not *GAPDH, 18S rRNA* or *MALAT1*, coprecipitated with ILF2 protein in human keratinocyte progenitors (Fig. 5d, e).

Using established methodologies[37–39], our quantitative estimations revealed that each human keratinocyte progenitor cell contains approximately 1.5-2 copies of *SNHG26* transcripts and about 104,125 ILF2 protein molecules (Supplementary Fig. 10e–g). During wound healing, *SNHG26* expression increases 27.5-fold (Fig. 1f), resulting in an estimated 41-55 copies of *SNHG26* RNA per wound keratinocyte cell. Notably, the stoichiometry of *SNHG26* RNA to ILF2 protein remains skewed, ranging from 1:1893 to 1:2540. In line with these findings, co-staining analysis showed *SNHG26* RNA and ILF2 protein co-localized within a condensate-like structure in the cell nucleus, but the abundance of ILF2 proteins greatly exceeded that of *SNHG26* RNA (Supplementary Fig. 10h–i). Interesting, we observed that the stability of SNHG26 was significantly decreased after ILF2 knockdown (Supplementary Fig. 10j). This result indicates that ILF2, or the nuclear condensates it forms, is crucial for maintaining the stability of SNHG26.

To identify other protein components within these condensates, we employed a proximity-dependent biotin identification (BioID) assay, which label proteins in close proximity to ILF2 that fused to BioID2, and subsequently identifying them through mass spectrometry (MS)[40]. We successfully overexpressed ILF2-BioID2 fusion protein in human keratinocyte progenitors (Supplementary Fig. 11a). The MS identified 176 proteins with fold change ≥ 2 which may be involved in the IF2-interactome (Supplementary Data 6). We also excluded keratins from this list, as they are commonly regarded as contaminants in MS studies[36]. Gene enrichments analysis showed that the function of these proteins was related to skin development, cell-cell junction, and focal adhesion (Supplementary Fig. 11b–e). Furthermore, to identify the components in the SNHG26-ILF2 nuclear condensates, we also overlapped these proteins with the ones co-purified with SNHG26 in the RNA pull-down assay, identifying seven proteins in common and among them two proteins-ACTB, ANXA2 reside in the nucleus (Supplementary Fig. 11g). Therefore, our results indicated that these two proteins may be the additional components of ILF2/SHG26 condensates.

ILF2 is a constitutively expressed nuclear protein that interacts with chromatin[41] and is a TF for both mitotic and inflammatory genes[42]. It is highly expressed in the human epidermis, as shown in the Human Protein Atlas database[43] (Supplementary Fig. 12a). *ILF2* expression was upregulated in the inflammatory phase (NW1) of human skin wound healing (Fig. 5f, Supplementary Fig. 12b). To study the effects of *SNHG26* knockdown on *ILF2* genomic occupancy, we performed ILF2 chromatin immunoprecipitation (ChIP)-sequencing in human keratinocyte progenitors. We identified 664 peaks (*p* value < 0.05) enriched

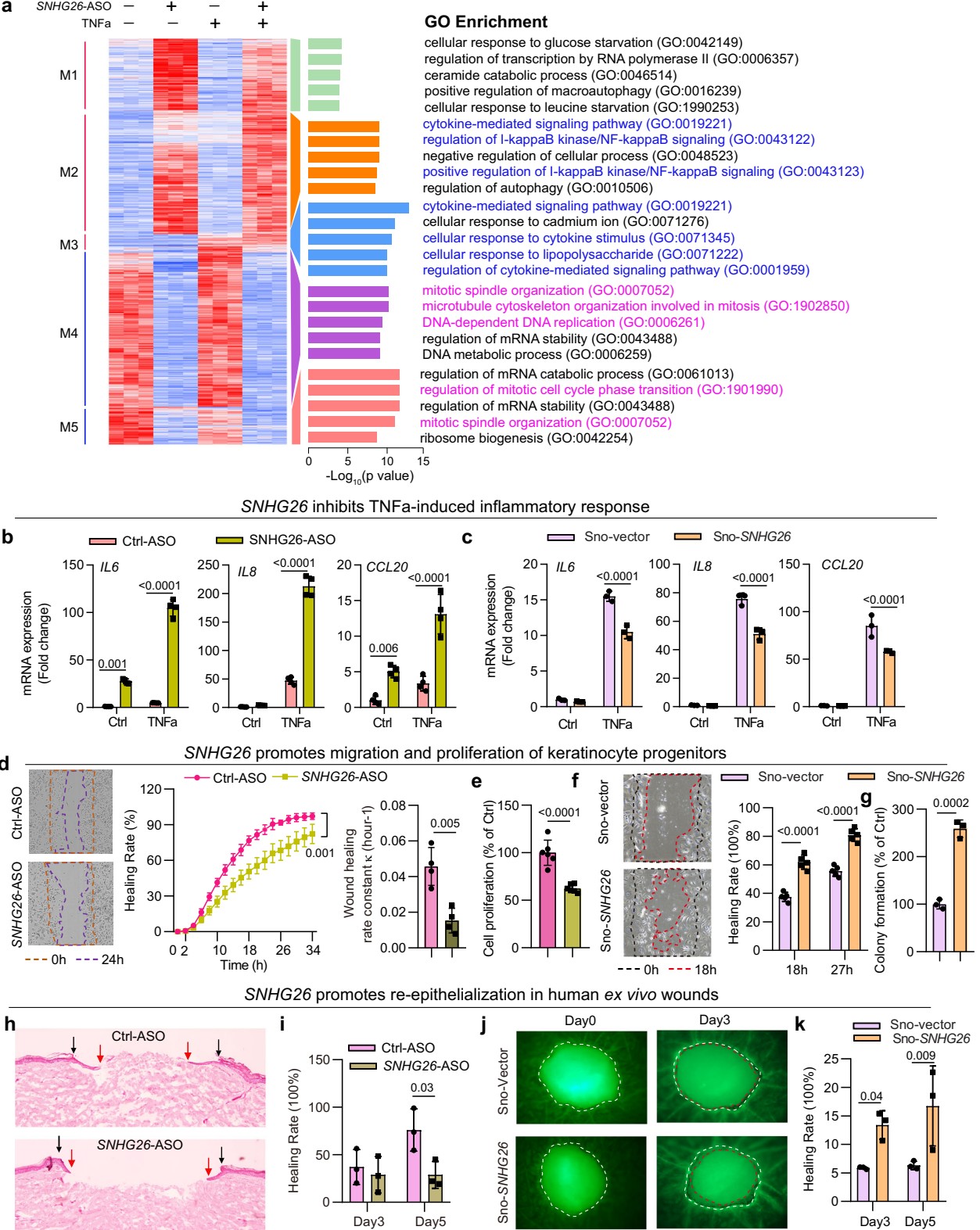

*SNHG26* inhibits TNFa-induced inflammatory response

*SNHG26* promotes migration and proliferation of keratinocyte progenitors

*SNHG26* promotes re-epithelialization in human *ex vivo* wounds

in keratinocytes treated with Ctrl-ASO and 589 peaks in cells with *SNHG26*-ASO (Fig. 5g, Supplementary Data 7). Focusing on ILF2 binding sites near gene promoters, we found that multiple cell growth-related pathways (e.g., EGFR1 pathway, DNA replication pre-initiation, G2/M transition, response to epidermal growth factor) were enriched in the Ctrl-ASO group. This indicates that ILF2 preferentially binds to cell growth-related gene loci in the presence of SNHG26 in

keratinocytes (Supplementary Fig. 12d). However, when SNHG26 was knocked down, ILF2 left these cell growth-related gene loci (Fig. 5g). We found that MAPK signaling and cellular response to cytokine stimulus were enriched in the ILF2 binding site-related genes in cells with *SNHG26*-ASO (Supplementary Fig. 12d).

Particularly noteworthy is the prominent ILF2 binding locus at the *JUN* promoter, located within a 1-kilobase proximity to the TSS, which

**Fig. 4 | SNHG26 inhibits the human keratinocyte inflammatory response and promotes re-epithelialization. a** Transcriptome profiling of human keratinocyte progenitors with *SNHG26* knocked down followed by TNFα treatment. GO analysis was performed for genes in each expression module (right). **b, c** qRT–PCR analysis of *IL6*, *IL8*, and *CCL20* expression in human keratinocyte progenitors after *SNHG26* knockdown (*n* = 4) (**b**) or overexpression (*n* = 3) (**c**) followed by TNFα treatment. **d, e** Scratch wound assay (*n* = 4) (**d**) and CyQUANT cell proliferation assay (*n* = 6) (**e**) of human keratinocyte progenitors with *SNHG26* knockdown. Wound healing rate constant κ (hour−1) in (**d**) was calculated using a one-phase decay model in GraphPad. **f, g** Scratch wound assay (*n* = 5) (**f**) and colony formation assay (*n* = 3) (**g**) of human keratinocyte progenitors with *SNHG26* overexpression. **h, i** Hematoxylin and eosin staining and quantification of human ex vivo wounds with topical application of *SNHG26*-ASOs (*n* = 3). Black arrows: the initial wound edges. Red arrows: newly formed epidermis in the front (day 5). Scale bar, 200 μm. **j, k** Representative images (top view) of ex vivo wounds stained with CellTracker™ Green CMFDA Dye (**j**) and quantification (*n* = 3) (**k**). The wounds were topically treated with *SNHG26* overexpression plasmids. Dashed white lines indicate the initial wound edges, and dashed red lines indicate newly formed epidermis. Data are shown as mean ± SD from two to three independent experiments (**b**–**k**). The data were analyzed by two-sided Fisher's exact test (**a**), two-way ANOVA analysis (**b**–**d**, **f**, **i**, and **k**), or two-tailed Student's t test (**e** and **g**).

ranked as one of the top binding sites in cells with *SNHG26* knockdown (Fig. 5g, Supplementary Fig. 13a, Supplementary Data 7). This result was further corroborated through ILF2 ChIP-qPCR analysis (Fig. 5h). Additionally, a ChIP–qPCR analysis revealed that TNFα treatment triggered ILF2 binding to the *JUN* promoter within 30 minutes (Fig. 5i). However, knockdown of *ILF2* completely abolished TNFα-induced *JUN* expression, emphasizing the crucial role played by ILF2 in driving *JUN* transcription (Fig. 5j, k, Supplementary Fig. 13b).

Jun/AP1 is an immediate-early gene and a master TF that controls the expression of a wide range of inflammatory genes in epidermal cells[44]. Upon examining the scRNA-seq data of mouse wounds, we found that not only was Jun expression increased (Fig. 5l), but the genes it regulated were also enriched among the upregulated genes in the wound-edge basal keratinocytes of the *Snhg26*-KO mice (Fig. 5m). Moreover, *Jun* emerged as a central hub connected to multiple genes involved inflammatory signaling, e.g., *Rel*, *Nfkbia*, *Mapk9*, and *Map3k8*[45–47] in wound keratinocytes of *Snhg26*-KO mice (Supplementary Fig. 13c). Additionally, in the mouse wound-edge epidermis (Fig. 2c, d), c-*Jun* was identified as the *Snhg26*-regulated TF that controls the most genes with aberrant expression after *Snhg26*-KO (Fig. 5n). It's worth noting that, unlike keratinocytes, *JUN* expression remained unaltered both in human fibroblasts transfected with *SNHG26*-ASO (Supplementary Fig. 13d) and in *Snhg26*-KO mice wounds (Supplementary Fig. 13e), indicating that *SNHG26* likely exerts distinct functional mechanisms in fibroblasts compared to keratinocytes. TNF-α also induced ILF2 binding to the promoters of other immediate early response genes, i.e., *IL6*, *IL8*, and *CCL20* (Fig. 5o). As a result, silencing *ILF2* led to a reduction in their expression in keratinocytes (Fig. 5p, Supplementary Fig. 13f). Intriguingly, *SNHG26* knockdown significantly increased the binding of ILF2 to the promoters of *IL6*, *IL8*, and *CCL20* (Fig. 5q).

Thus, we concluded that *SNHG26* interacts with ILF2 and prevents it from binding to inflammatory genomic loci, including the master TF JUN and several key cytokines and chemokines, consequently suppressing the inflammatory response in keratinocyte progenitors. Perturbation of SNHG26 impacts ILF2 binding site dynamics, shifting the focus from cell growth to inflammation-related processes within keratinocytes.

## SNHG26 directs ILF2 protein to the LAMB3 genomic locus

As *SNHG26* block the interaction of ILF2 with the promoters of inflammatory genes, we asked whether *SNHG26* may interact with chromatin and translocate ILF2 to another genomic locus. To assess this possibility, we map *SNHG26* occupancy genome wide by Chromatin Isolation by RNA Purification sequencing (ChIRP-seq) in human keratinocyte progenitors[48]. This method allowed us to isolate approximately 22% of *SNHG26* RNA in human keratinocyte progenitors (Fig. 6a, b). Importantly, the *SNHG26* probes did not retrieve *GAPDH*, nor did the LacZ probes retrieve *SNHG26*, confirming the method's specificity (Fig. 6b). Through ChIRP-seq, we identified 43 *SNHG26* occupancy sites across the genome (FDR < 0.05) (Fig. 6c, Supplementary Data 8). These sites were notably enriched within genic regions, particularly in regions annotated as promoters and introns (Fig. 6d). Notably, one of the most prominent *SNHG26* RNA occupancy

sites was located within its own genic region, further confirming the specificity of this ChIRP experiment (Fig. 6c, Supplementary Fig. 14a). Further analysis of the ChIRP-seq data using Multiple EM for Motif Elicitation (MEME)[49] revealed that *SNHG26* exhibited a preference for binding to a GA-rich polypurine DNA motif (Fig. 6e), indicating that *SNHG26* accesses the genome through specific DNA sequences.

We focused on the *LAMB3* gene due to its high-ranking among the top *SNHG26* ChIRP-seq peaks (Fig. 6c, f). Silencing *SNHG26* reduced ILF2 binding to the *LAMB3* gene, indicating that *SNHG26* directs ILF2 to the *LAMB3* locus by interacting with chromatin (Fig. 6g). Additionally, we demonstrated that ILF2 was crucial for driving *LAMB3* expression, as silencing *ILF2* significantly decreased the *LAMB3* mRNA level in keratinocytes (Fig. 6h).

*LAMB3* encodes subunit beta 3 of Laminin 332, which is required for keratinocyte attachment to the basement membrane in the epidermis and crucial for keratinocyte proliferation and migration[50]. Consistent with previous research[50], silencing *LAMB3* significantly impaired the migration, proliferation, and long-term growth of human keratinocyte progenitors (Fig. 7a–e, Supplementary Fig. 14b). Furthermore, *LAMB3* expression increased at the proliferative phase of human skin wound healing (NW7), supporting its role in wound re-epithelialization (Fig. 7f). Our scRNA-seq analysis of mouse wounds indicated that *Lamb3* was primarily expressed in basal keratinocytes and upregulated in wounds compared to the skin (Fig. 7g,h). However, *Snhg26*-KO mice exhibited significantly lower *Lamb3* expression in wound-edge basal keratinocytes compared to WT mice (Fig. 7h–j). This finding aligns with the delayed re-epithelialization observed in *Snhg26*-KO mice (Fig. 2e, f). Moreover, we compared ILF2 ChIP-seq and SNHG26 ChIRP-seq results, to better understand the functional connection between SNHG26 and ILF2. We identified LAMB3, PLEC, LINC02386, and DISP3 as common genes bound by SNHG26 and ILF2 (Supplementary Fig. 14c). ILF2 ChIP-seq further show that SNHG26 KD reduces ILF2 binding to these genes. Interestingly, in addition to LAMB3, PLEC and DISP3 have also been reported to promote cell proliferation[51,52].

In light of this multifaceted evidence, we propose a mechanism for the nuclear lncRNA *SNHG26*: it interacts with ILF2 and redirects it from inflammatory genomic loci, such as *JUN*, *IL6*, *IL8*, and *CCL20*, to the genomic locus of *LAMB3*, reshaping the gene expression program to facilitate the transition of keratinocyte progenitors from an inflammatory to a proliferative state.

## Discussion

In this study, we assessed the in vivo expression changes of lncRNAs throughout the human skin wound healing process, identifying *SNHG26* as an evolutionarily conserved lncRNA induced by injury and crucial in driving the transition of keratinocyte progenitors from an inflammatory to a proliferative state (Supplementary Fig. 14d). Using a genetically engineered mouse model, we provide conclusive evidence regarding the in vivo role of *SNHG26* in wound repair. It's worth noting that only a limited number of lncRNA mouse models have been created to date, and few have been assessed for their impact on wound healing[53]. The deficiency of *Snhg26* led to a shift in the transcriptional

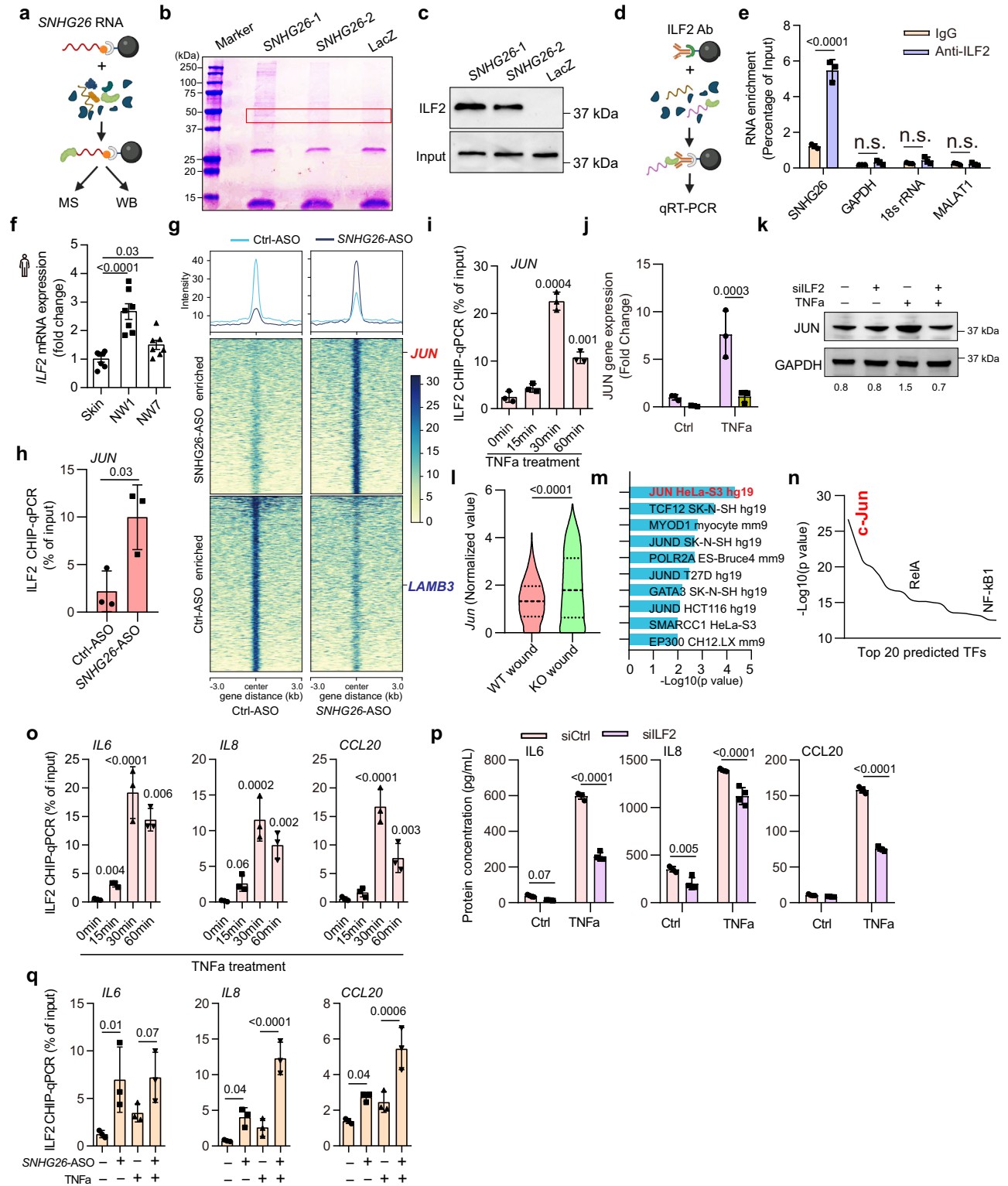

program, favoring the upregulation of inflammatory gene expression while simultaneously diminishing the expression of genes crucial for cell proliferation and migration in the basal keratinocytes at the wound edge. Furthermore, through experiments involving human keratinocyte progenitors and ex vivo wounds, we corroborated the conserved anti-inflammatory, pro-proliferative, and pro-migratory functions of *SNHG26*, confirming the clinical relevance of our findings.

In terms of mechanism, we have discovered that *SNHG26* interacts with the ILF2 protein, which forms a heterodimer complex with ILF3[42,54]. This complex plays a pivotal role in maintaining embryonic stem cell pluripotency and regulating gene expression related to the cell cycle[55]. Furthermore, it competes with Staufen-mediated mRNA decay, thus sustaining mitotic mRNA stability post-transcriptionally[54]. Additionally, the ILF2/ILF3 complex is known to bind to the promoters of numerous inflammatory genes (e.g., *IL2* and *IL13*) and immediate early genes (e.g., *EGR1*, *FOS*, and *JUN*), thereby driving their transcriptional activity[56–58]. Our study has unveiled that the injury-induced lncRNA *SNHG26* directs this multifunctional TF, ILF2, away from

**Fig. 5 | Silencing *SNHG26* increase binding of ILF2 to inflammatory genomic loci. a** Schematic illustration showing *SNHG26* pull-down assay. **b** SDS–PAGE protein bands: the highlighted protein bands were analyzed by MS. **c** WB analysis of ILF2 copurified with *SNHG26*. **d** Schematic illustration showing the RNA immunoprecipitation (RIP) assay. **e** ILF2 RIP followed by qRT–PCR analysis of copurified RNAs in human keratinocyte progenitors (*n* = 3). **f** qRT–PCR analysis of ILF2 expression in human acute wounds (*n* = 7). **g** Heatmap showing global ILF2 occupancy in *SNHG26* knockdown (SNHG26-ASO) and control (Ctrl-ASO) conditions. **h, i** ChIP–qPCR analysis of ILF2 at the promoter of *JUN* in human keratinocyte progenitors after *SNHG26* silencing (*n* = 3) (**h**) or treatment with TNFα for 0-60 minutes (*n* = 3) (**i**). **j, k** qRT–PCR analysis and western blot of *JUN* expression in human keratinocyte progenitors after *ILF2* silencing followed by TNFα stimulation (*n* = 3). **l** The expression of *JUN* in the basal keratinocytes of *Snhg26*-KO and WT mice wounds analyzed by scRNA-seq. **m** EnrichR analysis of the enrichment of TFs driving the upregulated genes in the wound-edge basal keratinocytes of *Snhg26*-KO

mice compared to that of WT mice. **n** TFs with overrepresented binding sites among the upregulated genes in *Snhg26*-KO mouse wound keratinocytes, as identified by MetaCore analysis. **o** ChIP–qPCR analysis of ILF2 at the promoters of inflammatory genes in human keratinocyte progenitors after TNFα stimulation (*n* = 3). **p** ELISA analysis of IL6, IL8, and CCL20 proteins produced by human keratinocyte progenitors with ILF2 silencing followed by TNFa stimulation (*n* = 3). **q** ChIP–qPCR analysis of ILF2 occupation at the promoters of inflammatory genes in human keratinocyte progenitors with *SNHG26* knockdown followed by TNFα stimulation (*n* = 3). Data are shown as mean ± SD from two to three independent experiments (**c**–**e**, **h**–**k**, **o**–**q**). The data were analyzed by one-way ANOVA (**f, i** and **o**), two-way ANOVA (**e, j, p** and **q**), two- sided Fisher's exact test (**m**) or two-tailed Student's t test (**h** and **l**). Panel a and d were created with BioRender.com released under a Creative Commons Attribution-NonCommercial-NoDerivs 4.0 International license (https://creativecommons.org/licenses/by-nc-nd/4.0/deed.en).

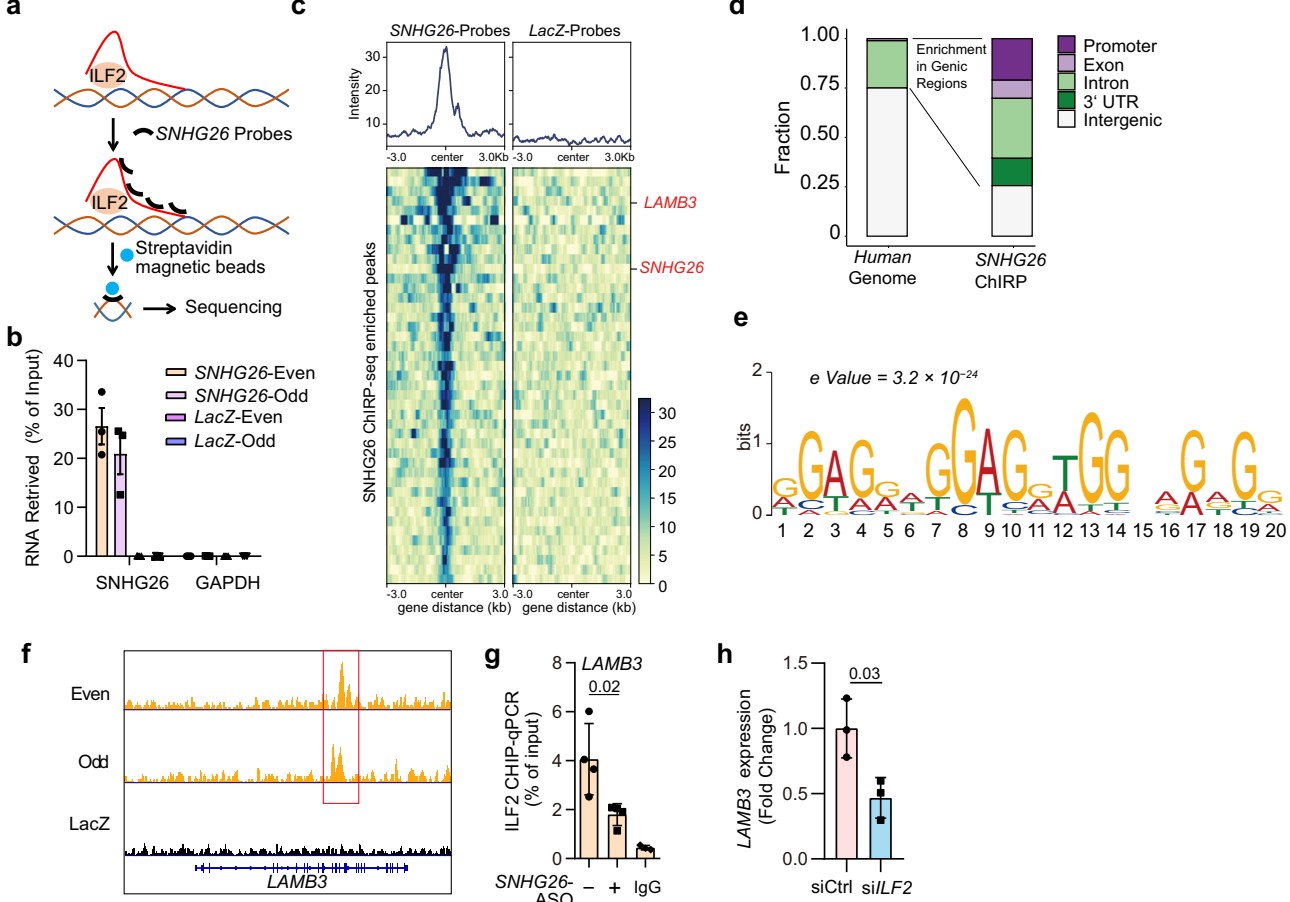

**Fig. 6 | *SNHG26* directs ILF2 protein to the *LAMB3* genomic locus. a** Schematic showing *SNHG26* ChIRP assay followed by DNA sequencing. **b** qPCR analysis of the percentage of *SNHG26* retrieved by its probes, *GAPDH* RNA was undetectable in the ChIRP assay (*n* = 3 biological replicates). **c** Heatmap showing global *SNHG26* ChIRP-seq signal in human keratinocyte progenitors. **d** *SNHG26* binding sites are enriched in genic regions, notably promoters and introns. **e** MEME analysis showing GA-rich homopurine motif enriched in *SNHG26* binding sites. **f** Genome browser tracks

showing *SNHG26* occupancy at the *LAMB3* locus. **g** ChIP–qPCR analysis of ILF2 at the promoter of *LAMB3* in human keratinocyte progenitors with SNHG26 silencing (*n* = 4). **h** qRT–PCR analysis of *LAMB3* expression in keratinocyte progenitors with *ILF2* silencing (*n* = 3). Data are shown as mean ± SD from one experiment (**b**) or two to three independent experiments (**g, h**). The data were analyzed by one-way ANOVA (**g**) or two-tailed Student's t test (**h**).

inflammatory genomic loci (such as *JUN, IL6, IL8,* and *CCL20*) and redirects it to the *LAMB3* gene, which is essential for cell proliferation and migration. This redirection enables keratinocyte progenitors to switch their cell states and thus perform different tasks at various stages of wound repair. Therefore, our study offers a fresh perspective on the established regulatory mechanisms governed by lncRNAs, shedding light on how they influence the functional specificity of transcription factors.

Based on research performed in the past two decades, lncRNAs have been proposed to enable a supragenomic layer of gene regulation that integrates genomic and epigenomic levels of gene expression[59]. Recently, Unlike protein regulators, lncRNAs operate swiftly without necessitating translation, and their levels can be dynamically adjusted due to their lower stability compared to proteins. We hypothesize that these features equip lncRNAs with the ability to regulate gene expression rapidly and precisely, which is

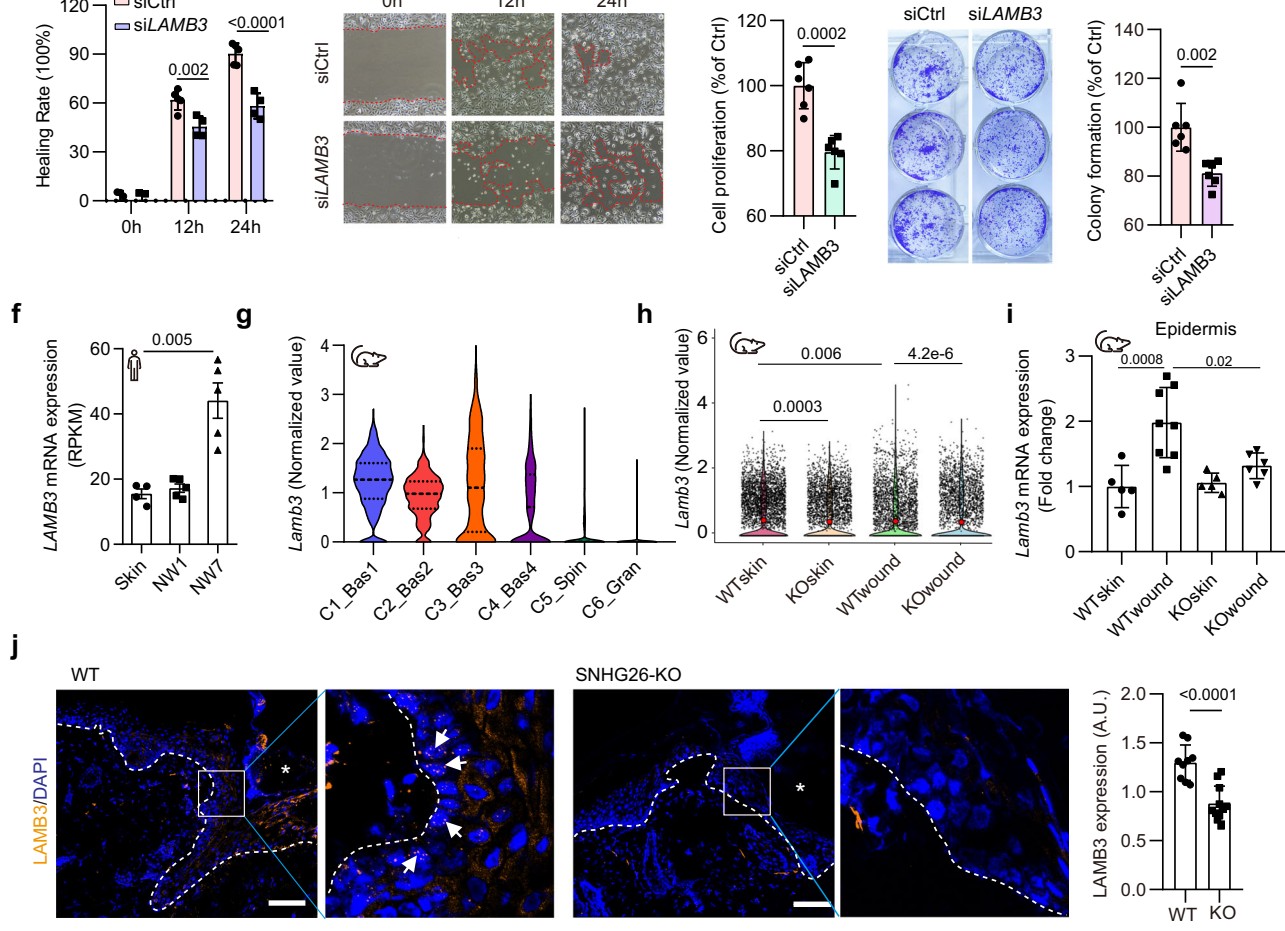

**Fig. 7 | LAMB3 promotes migration and proliferation of keratinocytes.**
**a**, **b** Scratch wound assay of human keratinocyte progenitors with *LAMB3* knock-down (*n* = 4) or control (*n* = 5) (**a**) and representative photos are shown in (**b**).
**c**–**e** CCK-8 cell proliferation assay (*n* = 6) **c** and colony formation assay (*n* = 6) **d**, **e** of human keratinocyte progenitors after *LAMB3* knockdown. **f** *LAMB3* expression in human acute wounds from 5 donors analyzed by RNA-seq (GSE174661). **g**, **h** Violin plots showing *Lamb3* expression in each cluster of keratinocytes (**g**) and in the *Snhg26*-KO mice vs. WT mice (**h**) as analyzed by scRNA-seq. **i** qRT–PCR analysis of

*Lamb3* expression in the skin and wound-edge epidermis of *Snhg26*-KO and WT mice (*n* = 5). **j** Immunofluorescence staining of LAMB3 in the wound-edge tissues of WT mice (*n* = 9) and SNHG26-KO mice (*n* = 11). The asterisk (*) indicates the wound edge. The arrow heads indicate the LAMB3+ basal cells. Basal layer keratinocyte expression of LAMB5 was quantified (right panel). Scale bar, 100 μm. Data are shown as mean ± SD from one experiment (**f**) or two to three independent experiments (**a**–**e**, **i**, **j**). The data were analyzed by one-way ANOVA (**f** and **h**), two-way ANOVA (**a**, **i**), or two-tailed Student's t test (**c**, **e**, **j**).

required for the swift transition of cell states during tissue repair. Consistent with this notion, our previous studies have shown that the lncRNA *WAKMAR2*[60] and *microRNA-132*[61] could also facilitate the inflammation–proliferation phase transition of keratinocyte progenitors. These findings open up exciting possibilities for harnessing non-coding RNAs with broad implications for regulating cell state transitions and further study is warranted to identify these physiologically relevant regulatory RNAs.

**Limitations of the study**
This study reveals a significant stoichiometric imbalance between *SNHG26* RNA and its protein binding partner ILF2, with ratios ranging from 1:1893 to 1:2540 in human keratinocyte progenitors. While the stoichiometric challenges of lncRNAs in relation to their targets and cofactors are acknowledged, the underlying mechanisms remain unclear[39,62]. Our co-staining analysis highlights the co-localization of *SNHG26* RNA and ILF2 proteins within a condensate-like structure in the cell nucleus. This prompts further investigation into whether biomolecular condensate formation may facilitate factor accumulation and contribute to achieving a more balanced stoichiometry between *SNHG26* and ILF2. Moreover, *SNHG26* exhibits broad expression across

various cell types. Our single-cell analysis using the *Snhg26* global KO mouse model has revealed significant gene expression alterations, not limited to keratinocytes but extending to other cell types such as fibroblasts and macrophages. Delving further into *SNHG26*'s functions in these diverse cell types will provide a comprehensive understanding of the pivotal roles played by this lncRNA.

## Methods
### Human samples
26 healthy volunteers were enrolled at the Karolinska University Hospital, Stockholm, Sweden. To study the expression patten of lncRNAs in human skin wound healing, we made 2-3 full-thickness wounds by using 3-mm biopsy punch at the skin of lower leg area or upper buttock area of each healthy donor. On day1, day7, or day30, the wound edge tissues were collected by using a 6-mm biopsy punch and snap-frozen for the following studies. Human skin samples used for ex vivo wound model were obtained from routine abdominal or breast reduction surgeries at the Department of Reconstructive Plastic Surgery, Karolinska University Hospital. Written informed consent was obtained from all donors for the collection and use of tissues for research. This study was approved

by the Stockholm Regional Ethics Committee and conducted according to the Declaration of Helsinki's principles.

## Mice

The *Snhg26* knockout mice and their control littermates used in this study were generated by Cyagen Biosciences Company (Santa Clara, US). The C57BL/6 J wild type mice used for *Snhg26*-ASO treatment were obtained from Charles River Laboratories (Sulzfeld, Germany). All the mice were breed under pathogen-free conditions in Comparative Medicine Biomedicum (KMB) animal facility at Karolinska Institutet under standard laboratory conditions, which included free access to food and water, a 12-hour light/dark cycle, a stable temperature range of 20-22 °C, and humidity maintained between 40-60%. The 8–10 weeks old mice with C57BL/6 J background were used in all the experiments. Both male and female mice were used. All the mouse experiments were approved by committee on animal experimentation of Swedish Board of Agriculture (Jordbruksverket).

## Mice treatment

The mice were wounded as previously described[63]. Briefly, all the mice were caged individually prior wound making. The back skin of 8-weeks old *Snhg26* knockout mice and their littermate controls at the telogen (resting) phase of the hair cycle were shaved with clippers followed by a treatment of depilatory cream. Two 4-mm full-thickness wounds extending through the panniculus carnosus were made on the dorsum on each side of midline by using biopsy punch. The mice were caged individually after surgical procedure to minimize the clawing from other mice. During the first two days, the animals received s.c. buprenorphine (0.03 mg/kg) twice a day for relieving pain and distress caused by the procedure. The wound area was photographed and measured every day until approximately 90% of wound area was healed. The wound area was quantified using ImageJ 1.49 software (National Institutes of Health) and adjusted for the area of the reference circle placed alongside the wounds. To investigate the healing rate in SNHG26-KO mice, we calculated the wound healing rate constant (day$^{-1}$) using a one-phase decay model in GraphPad, as described by Naik et al. [64].

To study the effect of knockdown of SNHG26 on mice wound healing, the SNHG26 Antisense LNA GapmeRs (SNHG26-ASO, 0.25nmol each wound) or Control GapmeRs (Ctrl-ASO, 0.25 mol each wound) were mix with MaxSuppressor™ In Vivo RNA-LANCEr II (Bioo Scientific) according to manufacturer's instruction in a total volume of 100 µl for each wound. The mixture was injected intradermally into 4 points around the wound-edges in the mice immediately after wounding.

To study the *Snhg26* expression change during wound healing in diabetic mice, two-4mm full thickness wounds were made on each of the db/db mice and WT mice as described above. The wound edge samples were collected on 3 days, 7 days, and 10 days after wounding. The expression of *Snhg26* was measured by qRT-PCR.

## Bulk RNA sequencing analysis of human skin and wound samples

Snap frozen skin and wound samples were homogenized by Tissue-Lyser LT (Qiagen) and total RNA was extracted by using miRNeasy mini Kits (Qiagen). The RNA quantity and integrity were determined by Nanodrop one (Thermo Fisher) and Agilent 2100 Bioanalyzer (Agilent Technologies), respectively. Long RNA sequencing libraries were generated to study the lncRNA expression.

Firstly, the ribosome RNA was depleted from 2 ug of total RNA by Epicentre Ribo-zero® rRNA Removal Kit (Epicentre). The RNA is then fragmented randomly in fragmentation buffer, followed by cDNA synthesis, and incorporating dUTPs in the second-strand synthesis step with NEB Next® Ultra™ Directional RNA Library Prep Kit for Illumina® (NEB). We used SPRI beads purification method to select the

insert size among 250bp-300bp (not including adapters) for library preparation. The libraries were sequenced on an Illumina Hiseq 4000 platform.

HISAT2 was used to align the reads to the human reference genome (GRCh38). The expression of lncRNAs is assessed by Cuffdiff and the raw count of each sample was normalized to Fragments Per Kilobase of transcript sequence per Millions value (FPKM). Only RNAs with median of FPKM ≥ 0.5 in all the samples were kept for the following analysis.

We process the file by a series of filter steps. Step1: Exon Number Filter: Filter the low expression level, low confidence single exon transcript, select the number of exon; Step2: Transcript Length Filter: Select transcript with length> 200 bp transcript; step3: Known Transcript Filter: filtering the transcripts which overlapped with database annotation exon region by Cuffcompare software, and add these annotation to lncRNA analysis; Step4: Coding Potential Filter: several popular software for coding potential analysis are adopted for coding potential filtering, including CPC[65], CNCI(vertebrate)[66], Pfam Analysis[67], and the predicted lncRNAs come from the intersection of these methods. The Multiple Experiment Viewer (MeV) software was used to perform the differential expression analysis between the sample groups (One way ANOVA, p < 0.005). Gene ontology (GO) and KEGG pathway enrichment analysis were performed by using the online tool EnrichR (https://maayanlab.cloud/Enrichr/)[68].

## Histological analysis

Paraffin-embedded tissue sections (5 µm in thickness) were deparaffinized and rehydrated, followed by hematoxylin and eosin (H&E) staining. To evaluate the epithelial tongue area of wound edge, eight sections from the wounds of four wild type mice and four *Snhg26*-KO mice were analyzed by Image J.

For immunofluorescence (IF), the sections were blocked with a blocking buffer (3% BSA, 0.5% Triton X-100 in PBS) for 30 minutes at room temperature. To check the macrophage infiltration, the rabbit anti-mouse CD68 antibody (1:10,000; MCA1957; AbD Serotec) was applied to the section for overnight at 4 °C. Matched IgG isotype controls were included for each staining. Primary antibodies were detected with Donkey anti-Rat IgG (H + L) Highly Cross-Adsorbed Secondary Antibody, Alexa Fluor™ 488 (1:500; A21208; Thermo Fisher). The detailed information of antibodies please see supplementary Supplementary Data 9. At last, the sections were mounted with ProLong™ Diamond Antifade Mountant with DAPI (P36971; Thermo Fisher). The results were visualized with LSM800 confocal laser scanning microscope (Carl Zeiss).

## Fluorescent in situ hybridization (FISH) and dual fluorescence staining

To visualize the expression of *SNHG26* in human skin and wound samples, the probes (C3 channel) for detecting human *SNHG26* RNAs were purchased from Advanced Cell Diagnostics and smRNA-FISH was performed using the RNAscope® Multiplex Fluorescent Detection Kit v2 (323100, Advanced Cell Diagnostics) by following manufacturer's instructions. Briefly, the fresh frozen tissue sections (8 µm) were fixed in cold 4% formaldehyde for 15 minutes followed by blocking with Hydrogen Peroxide for 10 minutes at room temperature. After digested with RNAscope® Protease IV at room temperature for 30 minutes, the sections were incubated with *SNHG26* probes for two hours at 40 °C in HybEZ™ II Hybridization System. Sequential hybridization procedures were performed with amplifiers and labeled probes (AMP1, AMP2, and AMP3). At last, TSA® plus fluorescein was assigned to C3 channel and mounted with ProLong™ Diamond Antifade Mountant with DAPI (P36971; Thermo Fisher). The results were visualized with LSM800 confocal laser scanning microscope (Carl Zeiss).

To visualize the expression of *SNHG26* RNA and ILF2 protein in human primary keratinocytes, sequential RNA protein detection was

performed by using RNA-Protein Co-detection Ancillary Kit (Advanced Cell Diagnostics, Cat: 323180) as described by the manufacture. We maintain human primary keratinocytes at early passages and sub-confluent (under 80% confluency) to prevent differentiation induced by confluency. Briefly, the cells were fixed with 4% formaldehyde for 15 minutes followed by target retrieval. The diluted ILF2 antibody (1:100) was add on the slide and incubated at 4 °C for overnight. After the second fixation with 4% formaldehyde and Protease Plus treatment, the sections were processed by RNAscope ® Multiplex Fluorescent staining protocol as described above. At last, the Alexa Fluor 488 conjugated secondary antibody was applied on the slide to visualize the ILF2 signal. The results were visualized with FV1000 confocal laser scanning microscope (Olympus).

### Spatial transcriptomics (ST)

The fresh frozen skin and wound tissues were embedded in Optimal Cutting Temperature compound (OCT, Sakura Tissue-TEK) on dry ice. The samples were then processed for ST analysis by using the Visium Spatial platform of 10x Genomics as per the manufacturer's instructions. Briefly, the H&E staining was first performed to assess the morphology and quality of the tissues. After permeabilization, reverse transcription and second-strand synthesis were performed on the slides. cDNA Library preparation, cleaning up, and indexing were then conducted. The pooled libraries were sequenced on NovaSeq6000 S4-200 (Illumina), generating ~300 M reads per section. The raw ST data were processed using the standard Space Ranger pipeline (version 1.2) with the GRCh38 human reference genome and GENCODE v38 gene annotations, and visualized by using BBrowser (BioTuring).

### Single cell RNA sequencing (scRNA-seq)

Mice skin and wound edge tissues were incubated with 5U/ml Dispase II at 4 °C for overnight. The epidermis was gently separated from dermis by using tweezers and incubated in 0.025% Trypsin/ EDTA (Thermo Fisher) for 10-15 minutes at 37 °C. The dermal cell suspension was prepared by using the whole skin dissociation kit (Miltenyi Biotec) according to the manufacturer's instructions. Equal amounts of epidermal and dermal cells were mixed. The red blood cells and dead cells were depleted by Red Blood Cell Removal Solution (Miltenyi Biotec) and Dead Cell Removal Kit (Miltenyi Biotec), respectively. 8000 cells were used to generate a scRNA-seq library with Chromium Next GEM Single Cell 3′ Reagent Kits v3.1 (10×Genomics) according to the manufacturer's protocol. The libraries were subjected to high-throughput sequencing on the Novaseq6000 S4-200 (Illumina) platform, generating 50k read pairs per cell.

Raw single-cell sequencing data were processed by using the standard 10× Cell Ranger (v5.0.1) analysis workflow, including demultiplexing, aligning to the GRCm39 (mm39) mouse reference genome, barcode counting, and unique molecular identifier (UMI) quantification. The doublets of cells predicted by Scrublet[69] were excluded. The filtered feature barcode matrices were used as input into a Seurat pipeline[70]. To this end, we removed mitochondrial genes, hemoglobin genes, ribosomal genes, genes expressed in less than 10 cells, as well as cells with less than 500 detected genes, less than 1000 UMIs and with more than 10% mitochondrial gene expression. Finally, 22,271 cells were retained for the subsequent analyzes, including 10,609 Snhg26-KO cells (5282 skin cells and 5327 wound cells) and 10,882 control cells (6281 skin cells and 4601 wound cells). The data were first normalized using the SCTransform[71] function. Principal component analysis (PCA) was carried out on the top 4000 variable genes among all the samples, and the first 40 PCs determined by ElbowPlot were used in the RunHarmony function[72] to mitigate potential batch effects among samples processed in different libraries. Uniform manifold approximation and projection (UMAP) plots were generated using the RunUMAP function with the first 40 harmonies. The clusters were obtained using the FindNeighbors and FindClusters functions with a resolution of 0.8. The

cluster markers were identified using the function of FindAllMarker. The cell types were annotated according to the overlaps between the cluster markers and well-known signature genes of each cell type from previous studies.

The migration, proliferation and inflammatory scores were calculated used Seurat's AddModuleScore function. The genes used for calculating the migration score were generated from GO:0051549. The genes for the proliferation score were sourced from a previously reported list of common proliferation markers (*ZWINT, E2F1, FEN1, FOXM1, H2AFZ, HMGB2, MCM2, MCM3, MCM4, MCM5, MCM6, MKI67, MYBL2, PCNA, PLK1, CCND1, AURKA, BUB1, TOP2A, TYMS, DEK, CCNB1,* and *CCNE1*)[73]. The genes for calculating the inflammation scores were from a summarized datasets from GO:0090594 and literature[8] (Supplementary Data 4). The CellChat package[28] was used to infer the potential cell-cell communications.

### Microarray analysis

Human adult primary keratinocytes (C0055C; Thermo Fisher) were cultured in Epilife medium supplement with Human Keratinocyte Growth Supplement (HKGS, S0015, Thermo Fisher) and 1X Penicillin-Streptomycin. When cells grew to 60% confluent, 10 nM *SNHG26*-ASO was transfected into the cells for 24 hours followed by TNFα stimulation for another 3 hours. The total RNA was extracted by using miR-Neasy mini Kits (Qiagen) following the manufacturer's instructions. The transcriptome profiling was performed by using Affymetrix Clariom S microarray in Bioinformatics and Expression Analysis (BEA) core facility at the Karolinska Institute. The raw.cel files were imported to Transcriptome Analysis Console (TAC, Thermo Fisher) for visualization. Normalized expression data (log2 transformed value) were exported from TAC and subjected to MeV software to analyze the differential expressed genes (DEGs) (One-way ANOVA, p < 0.001). The expression modules were defined by the hierarchical clustering on the DEGs. Gene ontology enrichment analysis in each module was performed by using the online tool EnrichR (https://maayanlab.cloud/Enrichr/)[68].

### Magnetic cell separation of different cell types from skin and wounds

After being washed in PBS, the skin and wound biopsies skin and wound biopsies were incubated with Dispase II (5 U/mL, ThermoFisher Scientific) at 4 °C for overnight. On the next day, the epidermis was separated from the dermis and digested in 0.025% Trypsin-EDTA at 37 °C for 15 minutes. The CD45+ and CD45- cells were isolated from the epidermal cell suspension by using CD45 microbeads (Miltenyi Biotec, Bergisch Gladbach, Germany). The dermis was processed with a Whole Skin Dissociation Kit according to the manufacturer's instructions (Miltenyi Biotec, Bergisch Gladbach, Germany). The dermis was digested with 12.5 μL of Enzyme P, 50 μL of Enzyme D and 2.5 μL of Enzyme A at 37 °C for 3 hours followed by processing with Medicon tissue disruptor (BD Biosciences, San Diego, CA). The fibroblasts, macrophages, and T cells were separated with CD90 + , CD68 + , and CD3+ beads from the dermal cell suspension.

### Gene deletion, silencing, and overexpression

For silencing of *SNHG26*, 10 nM LNA-long RNA-GapmeR-antisense-oligonucleotide (*SNHG26*-ASO or Ctrl-ASO) (Qiagen) was transfected into human primary keratinocytes by using Lipofectamine™ 3000 (ThermoFisher Scientific). For the depletion of *SNHG26* in Ker-CT human keratinocytes cell line (ATCC, Cat: CRL-4048), two sgRNAs targeting *SNHG26* loci were cloned into lentiCRISPRv2 plasmid, which was kindly provided by Prof. Yuping Lai and Prof. Dali Li in East China Normal University. The lentiCRISPRv2 plasmid was transfected into 293 T cells with the packaging plasmids pMD2G (AddGene) and psPAX2 (AddGene) for lentiviral production. Ker-CT cells were transduced with the virus supernatant for 12 hours and selected by 1 μg ml$^{-1}$

of puromycin. The expression of *SNHG26* was measured by qRT-PCR to confirm the *SNHG26* knockout.

For overexpression of *SNHG26* and *SNORD93*, the cDNA sequence of these two genes were synthesized and cloned into pcDNA3.1 or PZW1-SnoVector (Sangon, Shanghai). The keratinocytes at 60-70% confluence were transfected with these plasmids for 48 hours with Lipofectamine™ 3000 (ThermoFisher Scientific). The gene expression was measured by qRT-PCR.

### Quantification of *SNHG26* and ILF2 copy number

To quantify the *SNHG26* transcript, a standard curve by amplifying a dilution series of the PZW-Sno-*SNHG26* plasmid was created by using real-time PCR. Subsequently, we measured the Ct value of *SNHG26* in 50,000 keratinocytes via qRT-PCR and calculated the copy number based on the standard curve.

For the quantification of ILF2 protein, we conducted ILF2 Western blotting using a dilution series of ILF2 recombinant protein. The signal value of ILF2 was measured by Image J. A standard curve by plotting the signal values of the ILF2 immunoblot bands against the molarity of the ILF2 protein was then generated. The copy number of ILF protein in each keratinocyte was calculated by ILF2 band gray value and the standard curve.

### Scratch assay

Human adult primary keratinocytes were transfected with 10 nM *SNHG26*-ASO for 24 hours in Incucyte® Imagelock 96-well Plate (Sartorius). When cells grow to full confluent, the plate was placed in Essen® 96-well Wound Maker (Sartorius) to generate scratch wounds with approximately 800 μm width. After washing with PBS, EpiLife medium without any supplements was added to the cells. The plate was then placed into IncuCyte (Essen Bioscience) and imaged every two hours for 48 hours. The images in each time point were analyzed with the integrated metrics in the IncuCyte ZOOM software.

### Human ex vivo wound model

The human skin samples were obtained from abdominal reduction surgeries or mastectomies at the Department of Reconstructive Plastic Surgery, Karolinska University Hospital. The wound was made by using 2-mm biopsy punch on the surface of the skin. The injured skin was excised using an 8 mm biopsy punch and cultured in Dulbecco's Modified Eagle Medium (DMEM) supplemented with 10% fetal bovine serum (FBS) and antibiotics (penicillin 100 U/ml and streptomycin 100 mg/ml; 15140122; Thermo Fisher). 0.2 nmol *SNHG26*-ASO GapmeR or control oligos (Qiagen) were dissolved in 30% pluronic F-127 gel (Sigma-Aldrich) and topically applied on the wounds immediately after the injury and also three days later. The biopsies were collected at Day3 and Day5 after injury for histological analysis.

For the delivery of *SNHG26* overexpression plasmid DNA, we used in vivo-jetPEI (Cat. 201-10 G, Polyplus-transfection, Illkirch, France) reagent as per the manufacturer's guidelines. Specifically, 1.5 μg of plasmid DNA and 0.18 μl of in vivo-jetPEI were separately diluted in 2.5 μl of 5% glucose solution. These two dilutions were then combined, and after a 15-minute incubation at room temperature, 5 μl of this mixture was topically administered to the wounds immediately after injury, as well as two and four days later. The progress of wound healing was monitored by using CellTracker Green CMFDA fluorescent dye (Cat. C2925, Invitrogen, Waltham, MA)[74]. 4 μl of the dye solution (50 μM) was added to each wound tissue and allowed to incubate for 30 minutes at 37 °C with 5% CO_2. Following the incubation, the tissues were rinsed with PBS and imaged using a fluorescence microscope (Nikon eclipse Ni-E). The wound samples were imaged three and five days after the initial injury, and the percentage of re-epithelialization was quantified by using the Image J software. The re-epithelization of each wound was evaluated by the healing rate (%)=(Area within initial wound edges - Area within newly formed epidermis)*100/ Area within initial wound edges.

### RNA extraction and qRT-PCR

The tissue samples were homogenized by using TissueLyser LT (Qiagen) in Trizol (Thermo Fisher) prior to RNA isolation. For cells, Trizol was directly added into the cell culture plate and incubated at room temperature for 5 minutes. The total RNA was extracted following the instructions of the manufacturer of the Trizol reagent. The RNA was dissolved in nuclease-free water and the concentration was determined by Nanodrop one.

500 ng of total RNA was converted to cDNA using RevertAid First Strand cDNA Synthesis Kit (K1622, Thermo fisher). Quantitative PCR was performed by using SYBR™ Green PCR Master Mix (4309155, Applied Biosystems) or TaqMan™ Universal Master Mix II (4440038, Applied Biosystems). Target gene expression levels were normalized between samples to the internal control 18S rRNA in tissues and to GAPDH in cells. The fold change in gene expression was determined by the $2^{-\Delta\Delta Ct}$ method.

### RNA pulldown and mass spectrometry

*SNHG26* from human keratinocyte progenitors was amplified and inserted into pcDNA3.1(+) plasmid through double restriction enzyme digestion and ligation method. Linearized pcDNA3.1(+) vector (Thermo Fisher) expressing *SNHG26* and LacZ gene was used as a template to synthesize *SNHG26* and LacZ RNA by using MEGAscript™ T7 Transcription Kit (AM1334, Thermo Fisher). The RNA was purified by using MEGAclear™ Transcription Clean-Up Kit (AM1908, Thermo Fisher). After the concentration was measured by Nanodrop One, the biotin labeled RNA was generated by Pierce™ RNA 3′ End Desthiobiotinylation Kit (20163, Thermo Fisher). Biotin labeled RNA was further incubated with Streptavidin Magnetic Beads to form Beads-RNA complex. Each binding reaction was performed with 50 pmol of biotinylated RNA-beads and 200 ug of cell lysates prepared from human keratinocyte progenitors by using Pierce™ IP Lysis Buffer (87787, Thermo Fisher). After extensive washing, the proteins on the beads were eluted by heating samples at 96 °C for 10 minutes in 1X Laemmli buffer containing 1% SDS. The eluted proteins were subjected to SDS-PAGE analysis and Coomassie Blue Staining. A gel slice corresponding to 37-50 kDa protein bands from both SNHG26 and LacZ pulldowns were excised and subjected to mass spectrometry analysis.

### RNA Immunoprecipitation (RIP) assay

The RIP assay was performed as previously described[75]. Briefly, $2 \times 10^7$ human keratinocyte progenitors cell lysates were incubated with Pierce™ Protein A/G Magnetic Beads conjugated with mouse IgG or ILF2 antibody (sc-365283, Santa Cruz) at 4 °C for overnight. RNA-protein-antibody-Magnetic Beads complex was purified by Magnetic Separation. The immunoprecipitated RNAs were then extracted by Trizol. cDNA was synthesized using RevertAid First Strand cDNA Synthesis Kit (K1622, Thermo Fisher) and detected by qRT-PCR. Results were normalized to input RNA and shown as % enrichment.

### ChIP-PCR and ChIP-sequencing

The human keratinocyte progenitors were transfected with *SNHG26*-ASO or Ctrl-ASO. When grown to 90% confluent, the cells were crosslinked with 1% methanol-free formaldehyde (28906, Thermo Fisher) for 10 minutes at room temperature. The cross-linking was quenched with 125 mM glycine and incubated at room temperature for 5 minutes. The ChIP assay was performed by MAGnify™ Chromatin Immunoprecipitation System (492024, Thermo Fisher). The cell Lysis Buffer with Protease Inhibitors was added to the cell pellet and incubated on ice for 5 minutes. The cell lysate was sonicated using Bioruptor UCD-200 (Diagenode) with the setup of "30 sec ON and 30 sec OFF" at the highest voltage to generate 100-500 bp chromatin fragments. The sheared chromatin was diluted in a Dilution Buffer and incubated with Dynabeads® coupled ILF2 antibody and isotype control IgG at 4 °C for 2 hours. After extensive washing, the bead pellet was incubated with

Reverse Crosslinking Buffer with Proteinase K (20 mg/ml) at 55 °C for 15 minutes. After discarding the beads, the supernatant was further heated at 65 °C for 15 minutes. The genomic DNA was isolated by using the DNA Purification Magnetic Beads, and then fragmented randomly. After electrophoresis, DNA fragments of the desired length were gel purified. Adapter ligation and DNA cluster preparation were performed and subjected to DNBSEQ-G400 platform sequencing with PE100.

Raw sequencing data were processed by using Trimmomatic v0.36[76] to trim reads of low quality and shorter than 20 nucleotides. Clean reads were mapped to GRCh38 human reference genome using Bowtie2 (v2.3.5.1)[77] followed by removing PCR duplicates using Picard (v2.20.4) tools. The mapped reads were converted into bigwig format for IGV viewer[78] using deepTools bamCoverage function[79]. Peaks were called using MACS2 (v2.2.6) with the default parameter[80]. Peaks from two biological replicates were merged and filtered using Bedtools[81]. The peak regions of each group were compared and visualized using the computeMatrix and plotHeatmap functions inside the deepTools. The ChIP-purified DNA was also subjected to qPCR analysis using primers targeted the promoter region of indicated genes (Supplementary Data 9). The ChIP-qPCR results were shown as a percentage of enrichment relative to input DNA.

### ChIRP sequencing

The ChIRP sequencing was performed as previously described[48,82]. In this approach, we designed 28 3′-biotinylated complementary DNA oligonucleotides to span the entire length of *SNHG26* (3.1 kb). We used a similar set of probes targeting *LacZ* mRNA as a negative control. The *SNHG26* and *LacZ* probes were designed at Biosearch Technologies (www.biosearchtech.com) with the following criteria: number of probes = 1 probe /100 bp of RNA length; 2) percentage of GC = 45-50%; 3) Oligonucleotide length = 20. Sequences that had extensive complementarity to other sites in the genome or were repetitive were excluded. *LacZ*-specific probes were used as negative controls. To eliminate the artifacts due to the precipitation of non-specific DNA fragments from off-target hybridization of the probes, we ranked all probes based on their relative positions along the target RNA and split them into two pools. As the two sets of probes shared no overlapping sequences, the only target they have in common is the RNA of interest and its associated chromatin. To obtain high confidence ChIRP-seq signal, we performed two independent ChIRP-seq runs with "even" and "odd" probes separately and focused our analysis exclusively on the overlap between their signals. The DNA probes labeled with BiotinTEG at 3′ end were ordered from Eurofins Scientific. 3×10[7] human keratinocyte progenitors were firstly cross-linked by incubating with 1% formaldehyde and quenched with 125 mM glycine. After resuspending in the Swelling buffer, the nuclei were isolated and further cross-linked with 3% formaldehyde. The nuclei were resuspended in freshly supplemented Nuclear Lysis Buffer and sonicated for 3 hours at 4 °C with a Bioruptor. The chromatin was incubated with biotinylated probes pool at 37 °C for 4 hours in a hybridization oven. The RNA-Chromatin complex was captured with Dynabeads MyOne Streptavidin C1 magnetic beads. After extensive washing, the bead pellet was incubated with DNA Elution Buffer and the chromatin in the supernatant was reverse crosslinked with Proteinase K (20 mg/ml) at 50 °C for 45 minutes with gentle shaking. The DNA was purified by phenol-chloroform-isoamyl alcohol and subjected to NEBNext ChIP-seq library preparation. The paired-end sequencing libraries were sequenced by Illumina high-throughput sequencer (DNBSEQ PE100).

ChIRP sequencing analysis was performed by using the same pipeline of ChIP-seq as described above. We merged the peaks from two biological replicates and filtered out the ENCODE Blacklist[83] to construct the consensus peaks for each group using Bedtools[81]. The peaks were visualized using the computeMatrix and plotHeatmap functions inside the deepTools. The Motif Discovery function of MEME suite (version 5.5.4) was used to identify the de novo motif of *SNHG26* binding sites with the following parameters: the width of the expected motif was between 6 and 20 and the expected occurrence per sequence was set to zoops: zero or one site per sequence[49,84].

### BioID assay and mass spectrometry

MCS-BioID2-HA, kindly provided by Kyle Roux (Addgene plasmid # 74224), was used for the BioID assay as previously described in the literature[40,85,86]. ILF2 cDNA was synthesized and cloned into the MCS-BioID2-HA vector to produce the ILF2-BioID2 fusion protein. This recombinant plasmid, along with an empty vector, was transfected into human primary keratinocytes using Lipofectamine 3000 (Thermo Fisher), following the manufacturer's protocol. After 24 hours, the culture medium was supplemented with 50 μM biotin and incubation continued for another 16 hours. The cells were then washed with PBS and lysed with 540 μL of lysis buffer (8 M urea, 1 mM dithiothreitol (DTT), 1x protease inhibitor in 50 mM Tris·Cl) per 10-cm dish. The lysates were sonicated for 1 minute using an Ultrasonic Sample Processing System (LICHEN, Shanghai) at a 30% duty cycle and an output level of 4. After sonication, Triton X-100 was added to achieve a final concentration of 1%, and the lysate was further centrifuged to collect the supernatant. The supernatant was then incubated with 100 μL of Dynabeads™ MyOne™ Streptavidin beads (Thermo Fisher) overnight. Following extensive washing, proteins bound to the beads were eluted and analyzed by mass spectrometry.

### Quantification and statistical analysis

A minimum of four mice were included per experimental group. Mice were randomly allocated to different experimental group and male mice were used. For cell experiments, each groups contains a minimum of three independent biological replicates. For quantification of histological staining, sections from at least four mice were analyzed by ImageJ software. All data were expressed as mean ±standard deviation (SD) or mean ± standard error of the mean (SEM). Statistical significance between groups was determined using either a two-tailed Student's t-test or ANOVA analysis, by using GraphPad Prism 8 (GraphPad Software Inc, California, USA). For all statistical tests, *P* values < 0.05 were considered to be statistically significant.

### Reporting summary

Further information on research design is available in the Nature Portfolio Reporting Summary linked to this article.

### Data availability

The raw sequencing data have been have been deposited in the National Center for Biotechnology Information Gene Expression Omnibus (GEO) database under accession numbers: GSE174661 (human wound bulk RNA sequencing data), GSE216822 (microarray data of mice skin and wound epidermis), GSE216823 (microarray data of human keratinocytes transfected with SNHG26-ASO), GSE218430 (single-cell RNA sequencing data of mice skin and wounds, ILF2 ChIP-sequencing data and SNHG26 ChIRP sequencing data), and GSE241124 (human wound spatial transcriptomic data). The mass spectrometry raw data have been deposited in the Proteomics Identification Database under the accession number PXD055231 (SNHG26 pulldown followed by mass spectrometry) and PXD055235 (ILF2 BioID experiment). All software used in this study are listed in the Supplementary Data 9. Any additional information required to reanalyze the data reported in this work is available from the lead contact upon request. Source data are provided with this paper.

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

## Acknowledgements

We thank all the healthy donors and patients who took part in this study. We appreciate technical support from Dr. Beatrice Dyring-Andersen (University of Copenhagen) for quantifying the ILF2 protein copy numbers. We thank the microarray core facility at Novum, Bioinformatics and Expression Analysis, which is supported by the board of research at Karolinska Institutet and the research committee at the Karolinska Hospital. The computations/data handling was enabled by resources in projects of sens2020010 and SNIC2019/8-262 provided by the Swedish National Infrastructure for Computing (SNIC) at UPPMAX, partially funded by the Swedish Research Council through grant agreement no. 2018-05973. This work was supported by Swedish Research Council (Vetenskapsrådet) grants (2016-02051 and 2020-01400, N.X.L), Ragnar Söderbergs Foundation (M31/15, N.X.L), Welander and Finsens Foundation (Hudfonden, N.X.L.), Ming Wai Lau Centre for Reparative Medicine, LEO Foundation, Cancerfonden, Åke Wiberg Foundation, Karolinska Institutet, National Natural Science Fund for Excellent Young Scientists Fund of China (D.L), National Natural Science Foundation of China (82272294, D.L.), the Distinguished Medical Expert of Jiangsu Province (D.L.), Non-profit Central Research Institute Fund of Chinese Academy of Medical Sciences (2022-RC320-02, 2021-RC320-001, 2020-RC320-003), CAMS Innovation Fund for Medical Sciences (2021-I2M-1-059) and Jiangsu Provincial Medical Key Laboratory, Jiangsu Province Capability

Improvement Project through Science, Technology and Education (ZDXYS202204).

## Author contributions
D.L. and N.X.L. designed the research and wrote the manuscript. D.L., Z.L., L.Z., X.B., J.W., LH.L., Y.C., L.L., L.P., L.K., Y.X., J.W., X.Z., W.W., M.T., and M.P. performed the experiments. P.S. collected all the human samples. Z.L. LH.L. and D.L. performed bioinformatics analysis. All authors read and provided suggestions during manuscript preparation.

## Funding

## Competing interests
The authors declare no competing interests.
