## [Peer Review File · Nature Communications]

REVIEWER COMMENTS

Reviewer #1 (Remarks to the Author):

The paper by Li et al, titled “The lncRNA SNHG26 Drives the Inflammatory-to-Proliferative State Transition of Keratinocyte Progenitor Cells During Wound Healing,” presents a novel contribution to the understanding of wound healing mechanisms, particularly highlighting the role of long non-coding RNAs (lncRNAs). The authors employ a comprehensive screening approach to identify lncRNAs that are differentially expressed across different stages of wound healing, specifically inflammation-proliferation. Among these, small nucleolar RNA SNHG26 is pinpointed as a key regulatory factor during the transition of keratinocyte progenitor cells.

While the papers demonstrated how SNHG26 influences wound healing through a series of experiments, I have concerns about the level of significance of their findings. The foundation data for wound healing assay with the loss of function of SNHG26 show very limited defects in skin wound healing. It may show a slight delay in healing response, but the mechanism does not seem necessary to heal. This reduces the significance of the overall paper focusing on the mechanism of how SNHG26 can work. In addition, there are some suggestions and questions to enhance the clarity and depth of their description, as detailed below.

- 1- For lncRNA seq, the authors did not describe how they made size selection for RNA pool during library preparation as in methods. Moreover, the authors did not describe in methods the specific criteria that were considered for the selection of lncRNAs pool while analyzing the RNA seq data sets as its total RNA pool contains several different RNA species.
- 2- The authors' observation of elevated inflammatory cytokine expression and an increased CD68+ macrophage population in the dermal area of wounds in SNHG26 KO mice is intriguing. To further elucidate these findings, including a comparative analysis of CD68+ dermal macrophage patterns in mice treated with SNHG26- ASOs would be beneficial. Since ASOs seem to be ineffective in downregulating SNHG26 expression in the dermis, this comparison could provide valuable insights. Specifically, it would help determine whether the heightened immune responses observed in SNHG26 KO mice directly result from SNHG26 depletion or secondary feedback mechanisms.
- 3- Authors should refrain from stating the “inhibited wound repair in Snhg26-KO mice” as the results showed delayed wound repair.
- 4- In Figure 5A, the authors employed SNHG26-biotinylated RNA pull-down followed by mass spectrometry (MS) analysis to identify interacting protein partners and reported ILF2 as the sole identified protein. Given the high specificity and sensitivity of MS technology, it seems unusual to identify only a single protein in the respective band. To address this, it would benefit the authors to provide a more detailed explanation of this finding. This explanation could include insights into the

stringency of the pull-down conditions, the criteria used for protein identification in the MS analysis, and any steps taken to ensure the specificity of the interaction. Additionally, discussing whether other proteins were detected but not reported due to lower confidence levels or relevance could offer a clearer picture of the experimental outcomes.

5- In section “SNHG26 hijacks the transcription factor ILF2 from inflammatory genomic loci” while stating that co-staining analysis showed SNHG26 RNA and ILF2 protein the referenced figure number should be S6D, E.

6- The authors' observation of ILF2 and SNHG26 co-localizing in condensate-like structures within the nuclear compartment is an interesting finding. It would be insightful to investigate further aspects of these condensates especially during wound healing. Firstly, exploring whether the formation of these condensates contributes to the stability of SNHG26 would be valuable as it directly correlates its functional significance.

7- Moreover, identifying other proteins that are part of these condensates would be crucial. The composition of these complexes can offer significant insights into the specific transcriptional machinery and regulatory networks at play, for instance, in different stages of wound healing. Advanced techniques such as proximity labeling coupled with mass spectrometry could be employed to elucidate the proteomic landscape of these condensates.

Reviewer #2 (Remarks to the Author):

The authors investigated the role of the lncRNA SNHG26 in inflammatory-to-proliferative state transition of keratinocyte progenitors. They identify this lncRNA as the relevant one by profiling lncRNA expression changes during human skin wound healing. Using different experimental models (human wound, KO mice, ASO-silencing, human keratinocytes, 3D wound model) and different technologies (bulk RNAseq, single cell RNA-seq, CIRP, RNA-pool down and MS, etc) they demonstrated that SNHG26 promote inflammatory-to-proliferative state transition of keratinocyte.

Mechanistically, they proposed that SNHG26 interacted with ILF2 and relocated it from inflammatory genomic loci, such as JUN, IL6, IL8, and CCL20, to the genomic locus of LAMB3, important for cytoskeleton keratinocytes remodeling.

The article is very interesting. The authors have performed many experiments to prove their hypothesis, and the conclusions are mostly supported by the experimental data shown. The results shown could be very useful for regenerative medicine.

My main concern is that most of the results presented are shown at RNA level, this is specifically related to the mechanistic part. To strength the data they should demonstrate that the beta 3 subunit of laminin, Jun, ILF2 and the target cytokines chemokines, are indeed modulated upon SHHG26KO or silencing at protein level in the wound or in the different models. The mechanisms

identified would be further strengthened by combined FISH and immunofluorescent staining to visualize the special relationship between the lncRNA SNHG26 and the protein ILF2 in cells/tissue.

Additional points to clarify:

Fig1. SNHG26 expression is upregulated a NW7 (Fig1H) while its expression is barely detectable in normal skin. In human keratinocytes, however, seem to be expressed in normal conditions (fig1I and J). The authors should explain the difference of the expression level and give more detail about the keratinocyte progenitor conditions (proliferating/differentiating/confluency)

Fig2. The authors should better explain how they evaluated a 40% reduction in wound healing in SNHG26KO mice (fig2A). The healing rate is delayed, but why 40%.

Wound healing of Ctrl-ASO mice behave differently by WT mice (fig2a and Fig2b). The authors should explain this discrepancy. Is the injected ASO in control suitable? Is there inflammation going on?

Fig2. In fig2e the authors evaluated as markers for proliferation and differentiation Krt16 and Sprr1b, they also referred to references 21-23. However, these references do not refer at keratinocytes. I am not aware of a role of Sprr1b in keratinocytes migration. The authors should look at additional and specific markers for migration.

Fig5. The authors performed RNA pool down experiments. The authors should give more detail on the negative control they have used. In the method section they mention that they used the LacZ pool down. They should also show this experiment in fig 5b. Other bands of similar or higher intensity where pooled down (SNHG26-1), were these bands analyzed?

Fig5. In fig5e the authors should add an unrelated nuclear RNA as addition control for RIP?

Fig5. Fig5f shows expression of ILF2 at mRNA level. Does ILF2 expression profile follows the one of SNHG26 during wound healing also at protein level? Is ILF2 expression specific for keratinocytes?

The author also reported increased Jun transcription after siLF2 (fig5j), does Jun increase also at protein level?

Fig5. The authors try to use different approaches to demonstrate the link between SNHG26, Jun and ILF2. Most of the experiments are however carried on at transcriptional level, without showing the protein. For instance, in Fig 5 they measure a significant but small reduction of IL6, IL8 and CCL20 mRNA upon siILF2. Does this small reduction at mRNA level affect the level of these pro-inflammatory cytokines and chemokines?

Fig6. The authors focused their attention on the LAMB3 locus. Is the expression level of LAMB3 decreased in SNHG26KO mice wound?

To better demonstrate the mechanism they have identified, would be interesting to visualize by combined FISH and immunofluorescent staining the special relationship between the lncRNA SNHG26 and the protein ILF2 in one or more experimental model they have used.

Reviewer #3 (Remarks to the Author):

In this interesting report Li et al. study the function of the non-coding RNA SNHG26 during wound healing. The analysis of human skin samples during wound healing identifies several conserved lncRNAs including SNHG26 predicted to be a positive regulator of proliferation and migration and a negative regulator of genes involved in inflammation. Knocking out the entire SNHG26 genomic locus in mice they found wound healing impairment. The expression profile indicated that indeed SNHG26 is a positive regulator of migration and proliferation. The author also found an increase of macrophages during wound healing in the KO mice respect to controls. To understand this KO phenotype at cell type level they performed scRNA-seq. As this analysis pointed to Keratinocytes, the authors tested SNHG26 function with an ASO in human keratinocytes in vitro through transcriptome profiling and cell assays. With pull-down assay and RIP the authors showed the interaction between ILF2/NF45 and SNHG26. With Chip-seq they showed that ILF2/NF45 binding to the genome can be positively and negatively regulated by SNHG26. The authors also show the genomic chromatin binding of SNHG26 itself and identified that SNHG26 regulates the recruitment of ILF2 in the promoter of LAMB3, a well characterized gene involved in cell adhesion and migration. Unfortunately, there is a missing integration of the data between Chip-seq and ChIRP-seq to understand a broader functional connection between SNHG26 and ILF2.

Overall the manuscript contains very interesting results and a refreshing mix of omic data. However often the conclusions are not supported by the data.

Below several points that should be taken in account before publication:

“..The screening of LncRNAs identified a small nucleolar RNA host gene 26 (SNHG26)...” This sounds like something is wrong for readers as small nucleolar RNA and LncRNA are quite different categories. The explanation of the complex locus should be done in the introduction.

Relative to Fig1. The analysis of the human to mouse non coding RNAs conservation is generally tricky and a quick look of it for the human SNHG26 does not show an obvious result. This conservation analysis should be shown in supplementary data, including a sequence comparison of SNHG26

Fig1.D NW1, NW7 , NW30 in human and NW3, NW7, NW10 are assumed to be the right time points to be defined as inflammatory, proliferative and remodeling phases. However, the authors should show data in their settings to prove these associations.

“...Notably, the SNHG26 gene locus also contains a small nucleolar RNA SNORD93 in both human and mice (Figure 1C, Figure S1A, B), but only SNHG26 showed increased expression in keratinocytes during wound repair (Figure S1E, F)...” This is quite hard to understand. How is possible that the lncRNA (that contains the snoRNA) is differentially expressed while the contained snoRNA is not. Rt-pcr with one primer on the snoRNA and one in the remaining part of the lncRNA should help.

The strategy to generate the KO is not optimal. For instance it would be better to add a poli-A at the end of the first exon, to delete the expression of the lncRNA. Anyway, having removed the entire locus of the lncRNA might have impaired the regulation of surrounding genes if the lncRNA is working in “cis”. At least this should be checked.

How the ASO was used in vivo is not explained in the method. This is important as the author obtained a delivery specific to the epidermis and not the dermis (Figure S2F).

Fig2c,d GSEA support the impact on the proliferation and migration. However it is not possible to claim anything based on GO term enrichment as the authors show “gene number” instead of adjpV (or pV at least). So far the author can only claim an effect of the two validated genes IL6, IL1b.

The data to support the conclusions on exacerbated inflammatory response in KO are not enough. In Fig 2G there is a 2 fold enrichment of macrophages. However from the analysis provided on scRNA-seq is not possible to understand if these and other immune cells are increased in the the wound time point in KO vs wt. the normalization on Fig3D is wrong as the percentage should be done on the tola of cells and not on the toal cell number in each single cluster.

Fig3H,I The Data representation is confusing and, most importantly, does not allow for a statistical analysis.

Related to Fig3g,h,i – if SNHG26 is expressed equally on the different Basal clusters why the analysis is not performed on the union of these cell clusters?

An intersection between the migration/proliferation genes from the microarray data from 1 and scRNA-seq would be informative. The same is true for inflammatory genes.

From Method section - The genes for the migr/prolif score were taken from GO terms. Why inflammatory genes were not?

Related to Fig3g,h,i - From the analysis it seems that different basal cell populations do different thing during wound healing. The author should confirm this by histology the existence of this heterogeneity in skin .

Following this, I'm not sure if it is correct talk about “cell state transition” rather than something like to “activate cell population with different potencies”.

“This ASO was designed to target the exons of SNHG26, and it led to a significant reduction in SNHG26 expression while leaving SNORD93 unaffected (Figure S5A, B)” (row217), hard to understand how this is possible. Please explain.

Fig4j – it is har to understand how the quantification was performed as from the representative images it is not possible to see any difference.

The results of MS relative to Fig5B should be put in a supplementary table. Is the band in the red rectangle only visible in SNHG26-1?

Fig5.D Are the differences between IgG and Anti-ILF2 for gapdh and 18s significant?

Enrich R pathway analysis in FigS6G, H is important but not convincing. are there other GO term that are enriched? Are other GO terms enriched (including BP, CC, MF, Kegg terms)? MAPK signaling pathway is also a regulator of proliferation in keratinocytes. Interleukin2 pathway is in Ctlr-ASO and not in SNHG26.

“SNHG26 hijacks ILF2 from promoter” in Fig5N does not make any sense.

Fig5 contains interesting data however so far the conclusions proposed are not supported by the data. The data so far showed that SNHG26 interacts with ILF2, that the absence of SNHG26 can prevent or favor the ILF2 binding. ILF2 binding on Jun, IL6, IL8 and CCL20 promoters is negatively regulated somehow by SNHG26.

Are the loci selected in Fig.5P identified from the Chip-seq?

Row 275 the use of the word “enriched” is wrong.

The authors failed to cite Wu TH, et PLoS One. 2019 PMID: 31022259. Where it was shown that ILF2/NF45 targets transcriptionally Jun.

ChIRP-seq experiments seems suboptimal as only 43 genomic loci were found. This is an important issue as no intersection with the CHIP-seq in Fig 5 is shown (most likely as the intersection is null). The hypothesis that SNHG26 facilitate or favor somehow the binding of ILF2 in different loci seems true. However it is hard to believe that SNHG26-dependent ILF2 localization on LAMB3 is an important mechanism rather than one of the few genes identified by ChIRP-seq regulating.

Related to S6N - A general weak point of this manuscript is that the KO was obtained with the deletion of the entire locus of SNHG26. However it is clear that SNHG26 works also in “cis” on his own locus (the best peak in ChIRP-seq?). This type of regulatory regions can influence the expression of neighboring genes.

The statistics are present most of the times. However it is necessary to specify the number of replicates.

REVIEWER COMMENTS

RE: Nature Communications manuscript NCOMMS-23-63676

TITLE: The lncRNA SNHG26 drives the inflammatory-to-proliferative state transition of keratinocyte progenitor cells during wound healing

Point-by-point response:

Reviewer #1:

The paper by Li et al., titled "The lncRNA SNHG26 Drives the Inflammatory-to-Proliferative State Transition of Keratinocyte Progenitor Cells During Wound Healing," **presents a novel contribution to the understanding of wound healing mechanisms, particularly highlighting the role of long non-coding RNAs (lncRNAs)**. The authors employ a comprehensive screening approach to identify lncRNAs that are differentially expressed across different stages of wound healing, specifically inflammation-proliferation. Among these, small nucleolar RNA SNHG26 is pinpointed as a key regulatory factor during the transition of keratinocyte progenitor cells.

While the papers demonstrated how SNHG26 influences wound healing through a series of experiments, I have concerns about the level of significance of their findings. The foundation data for wound healing assay with the loss of function of SNHG26 show very limited defects in skin wound healing. It may show a slight delay in healing response, but the mechanism does not seem necessary to heal. This reduces the significance of the overall paper focusing on the mechanism of how SNHG26 can work. In addition, there are some suggestions and questions to enhance the clarity and depth of their description, as detailed below.

R1. We appreciate the reviewer's insights regarding the significance of our findings on SNHG26. Our *in vivo* investigations revealed that SNHG26 gene ablation (KO) and transient inhibition (Snhg26-ASO injection) resulted in significant delays in wound closure rates of approximately 40% and 37.5%, respectively (**Fig. 2A, B**). These effects are comparable to or greater than those observed in similar studies^{1,2}. Also, our investigations consistently revealed a clear molecular phenotype underlying the delayed wound healing (**Fig. 2-3, Fig. S4-S8**).

In murine wound healing models, it is challenging to block wound healing completely due to their robust and redundant repair mechanisms³. Additionally, murine models cannot fully mimic the complexity of human chronic non-healing wounds⁴. Therefore, it is rare to show any mechanism is necessary for healing, as its intervention entirely blocks healing. A clear delay in wound healing, as observed in our study, is sufficient to highlight the important role of SNHG26 in this process.

Even subtle delays can be biologically significant, particularly under stress or comorbid conditions, potentially leading to chronic wounds in clinical contexts. In line with this, we found

a significant reduction in SNHG26 expression in human diabetic foot ulcers and in wounds of diabetic mice (db/db) compared to controls (**Fig. S3J-L**). This highlights the potential implications of our findings in chronic wound pathology.

Taken together, our study employs a genetic mouse model, the gold standard for linking genotype to phenotype in lncRNA research⁵, along with human keratinocyte progenitor cells and human *ex vivo* wound models. This comprehensive approach provides definitive evidence supporting the functional significance of SNHG26 in regulating epidermal stem cell activity during wound repair.

We are immensely thankful to Reviewer #1 for the suggestions to enhance the clarity and depth of this manuscript. During the revision, we have undertaken substantial new experiments and analyses. These include CD68 immunostaining on murine wounds treated with SNHG26-ASO (**new Fig. S5E**), analysis of SNHG26 stability in human keratinocytes with ILF2 knockdown (**new Fig. S10J**), and a proximity-dependent biotin identification (BioID) assay followed by mass spectrometry (MS) analysis to identify additional protein components in the SNHG26-ILF2 complex (**new Fig. S11**). Additionally, we have clarified the methods of analyzing lncRNA seq and provided a more detailed explanation of the SNHG26-biotinylated RNA-MS results (**Table S5**). These enhancements have significantly solidified the mechanism we propose.

1.1 For lncRNA seq, the authors did not describe how they made size selection for RNA pool during library preparation as in methods. Moreover, the authors did not describe in methods the specific criteria that were considered for the selection of lncRNAs pool while analyzing the RNA seq data sets as its total RNA pool contains several different RNA species.

R 1.1: Per reviewer's suggestions, we have included this information in the revised methods section (under the subtitle 'Bulk RNA sequencing analysis of human skin and wound samples'), which is copied as below:

"Firstly, the ribosome RNA was depleted from 2 ug of total RNA by Epicentre Ribo-zero® rRNA Removal Kit (Epicentre). The mRNA is then fragmented randomly in a fragmentation buffer, followed by cDNA synthesis and incorporating dUTPs in the second-strand synthesis step with NEB Next® UltraTM Directional RNA Library Prep Kit for Illumina® (NEB). We used SPRI beads purification method to select the insert size between 250bp-300bp (not including adapters) for library preparation. The libraries were sequenced on an Illumina Hiseq 4000 platform.

HISAT2 was used to align the reads to the human reference genome (GRCh38). The expression of lncRNAs is assessed by Cuffdiff and the raw count of each sample was normalized to Fragments Per Kilobase of transcript sequence per Millions value (FPKM). Only RNAs with a median of FPKM \geq 0.5 in all the samples were kept for the following analysis.

We process the file using a series of filter steps. Step1: Exon Number Filter: Filter the low

expression level, low confidence single exon transcript, select the number of exons; Step2: Transcript Length Filter: Select transcript with length > 200bp transcript; step3: Known Transcript Filter: filtering the transcripts which overlapped with database annotation exon region by Cuffcompare software, and add these annotations to lncRNA analysis; Step4: Coding Potential Filter: several popular software for coding potential analysis are adopted for coding potential filtering, including CPC, CNCI(vertebrate), Pfam Analysis, and the predicted lncRNAs come from the intersection of these methods. The Multiple Experiment Viewer (MeV) software was used to perform the differential expression analysis between the sample groups (One-way ANOVA, $p < 0.005$). Gene ontology (GO) and KEGG pathway enrichment analysis were performed by using the online tool EnrichR (<https://maayanlab.cloud/Enrichr/>).

1.2 The authors' observation of elevated inflammatory cytokine expression and an increased CD68+ macrophage population in the dermal area of wounds in SNHG26 KO mice is intriguing. To further elucidate these findings, including a comparative analysis of CD68+ dermal macrophage patterns in mice treated with SNHG26- ASOs would be beneficial. Since ASOs seem to be ineffective in downregulating SNHG26 expression in the dermis, this comparison could provide valuable insights. Specifically, it would help determine whether the heightened immune responses observed in SNHG26 KO mice directly result from SNHG26 depletion or secondary feedback mechanisms.

R.1.2: We appreciate the excellent suggestion from the reviewer. Following the recommendation, we performed CD68 immunostaining on wounds treated with SNHG26-ASO or Ctrl-ASO. Consistent with the results from Snhg26-KO mice, the number of CD68+ macrophages was significantly increased in wounds treated with SNHG26-ASO (**new Fig. S5E**). These data underscore that the enhanced immune response observed in SNHG26-KO mainly results from the epidermal depletion of SNHG26.

For reading convenience, we copied the results here:

Figure S5E. Immunofluorescence analysis of CD68+ macrophages in mouse wounds injected with SNHG26-ASO or Ctrl-ASO ($n=3$). Scale bar, 20 μ m.

1.3 Authors should refrain from stating the "inhibited wound repair in *Snhg26*-KO mice" as the results showed delayed wound repair.

R.1.3: As suggested by the reviewer, we have rephrased the statement as "delayed wound repair in *Snhg26*-KO mice" (see **page 10, line 183-184**).

1.4 In Figure 5A, the authors employed SNHG26-biotinylated RNA pull-down followed by mass spectrometry (MS) analysis to identify interacting protein partners and reported ILF2 as the sole identified protein. Given the high specificity and sensitivity of MS technology, it seems unusual to identify only a single protein in the respective band. To address this, it would benefit the authors to provide a more detailed explanation of this finding. This explanation could include insights into the stringency of the pull-down conditions, the criteria used for protein identification in the MS analysis, and any steps taken to ensure the specificity of the interaction. Additionally, discussing whether other proteins were detected but not reported due to lower confidence levels or relevance could offer a clearer picture of the experimental outcomes.

R.1.4: We appreciate the valuable suggestions from the reviewer. Our MS experiment identified 38 proteins with over a 2-fold increase in the SNHG26 group compared to the LacZ control group (**new Table S5**). We excluded keratins from this list, as they are commonly considered contaminants in MS studies⁶. Given that SNHG26 is primarily expressed in the nucleus, we focused on nuclear proteins, particularly ILF2, which is among the top3 abundant proteins by SNHG26 pulldown. ILF2 is a nuclear transcription factor which is known to play critical roles in cell proliferation and the inflammatory response^{7, 8}. We have detailed this in the results section (**Page 15, lines 284-290**). For reading convenience, we copied the new Table S5 here:

Accession Number	prot_matches_sig +1 (SNHG26+1)	prot_matches_sig +1 (NC+1)	Fold change	Subcellular Location
ANXA2_HUMAN	18	1	18	Cytosol, Nucleus
IF2G_HUMAN	13	1	13	Intracellular
ILF2_HUMAN	13	1	13	Nucleus
EIF3E_HUMAN	24	2	12	Cytosol
OST48_HUMAN	11	1	11	Endoplasmic reticulum
HNRPQ_HUMAN	11	1	11	Cytosol
SYRC_HUMAN	10	1	10	No information.
PTBP1_HUMAN	10	1	10	Nucleoplasm
TSP1_HUMAN	9	1	9	Plasma membrane
ACOT9_HUMAN	9	1	9	Cytosol , Nucleus
HNRPK_HUMAN	9	1	9	Nucleus

AP3M1_HUMAN	8	1	8	Intracellular
DAZP1_HUMAN	8	1	8	DAZAP1,Nucleus , Cytosol
NEUA_HUMAN	8	1	8	Cytosol, Nucleus
TBL2_HUMAN	8	1	8	Cytosol
PTBP3_HUMAN	8	1	8	Nucleoplasm
FEN1_HUMAN	7	1	7	Nucleoplasm , Nucleoli
DEK_HUMAN	7	1	7	Nucleoplasm
HNRPD_HUMAN	7	1	7	Nucleoplasm
RBMS2_HUMAN	7	1	7	Nucleus, Cytosol
PCBP2_HUMAN	6	1	6	Nucleoplasm, Cytosol
SYEP_HUMAN	6	1	6	Cytosol
SYIC_HUMAN	6	1	6	Cytosol
SYK_HUMAN	6	1	6	Plasma membrane, Cytosol
EIF3G_HUMAN	6	1	6	Cytosol
RRBP1_HUMAN	6	1	6	Endoplasmic reticulum
EF1G_HUMAN	39	7	5.6	Cytosol
LAMB3_HUMAN	9	2	4.5	Cytosol
PA2G4_HUMAN	22	5	4.4	Cytosol
IF2B_HUMAN	18	5	3.6	Cytosol, Nucleoli
IF4A1_HUMAN	18	5	3.6	Cytosol
EF1A3_HUMAN	48	17	2.8	Cytosol
ACTB_HUMAN	23	9	2.6	Cytosol, Nucleus
EIF3F_HUMAN	15	6	2.5	Cytosol
ALBU_HUMAN	9	4	2.2	Cytosol
THIO_HUMAN	195	87	2.2	Peroxisomes
LAMC2_HUMAN	6	3	2	Cytosol
LAMA3_HUMAN	6	3	2	Cytosol

1.5 In section "SNHG26 hijacks the transcription factor ILF2 from inflammatory genomic loci" while stating that co-staining analysis showed SNHG26 RNA and ILF2 protein the referenced figure number should be S6D, E.

R.1.5: We thank the reviewer for pointing this out. We have revised the text accordingly.

1.6 The authors' observation of ILF2 and SNHG26 co-localizing in condensate-like structures within the nuclear compartment is an interesting finding. It would be insightful to investigate

further aspects of these condensates especially during wound healing. Firstly, exploring whether the formation of these condensates contributes to the stability of SNHG26 would be valuable as it directly correlates its functional significance.

R.1.6: We greatly appreciate the reviewer's suggestions. We silenced ILF2 in keratinocyte progenitors and treated them with actinomycin-D (5 $\mu\text{g}/\text{mL}$) to block RNA transcription⁹. We observed a significant decrease in SNHG26 stability after ILF2 knockdown (**new Fig. S10J**), indicating that the interaction between ILF2 and SNHG26 is crucial for maintaining SNHG26 stability. For convenience, we have included the results here and revised the result section on **page 16, line 301-304**.

Fig. S10J

Figure S10J. Keratinocytes were transfected with ILF2 siRNA and then treated with actinomycin D (5 $\mu\text{g}/\text{mL}$) for 2-8 hours. SNHG26 expression was measured by qRT-PCR ($n=3$).

1.7 Moreover, identifying other proteins that are part of these condensates would be crucial. The composition of these complexes can offer significant insights into the specific transcriptional machinery and regulatory networks at play, for instance, in different stages of wound healing. Advanced techniques such as proximity labeling coupled with mass spectrometry could be employed to elucidate the proteomic landscape of these condensates.

R.1.7: We fully agree with the reviewer. To identify other protein components within these condensates, we used a proximity-dependent biotin identification (BioID) assay. This assay labels proteins near ILF2 fused to BioID2, which are then identified through mass spectrometry (MS)¹⁰. We successfully overexpressed ILF2-BioID2 fusion protein in human keratinocyte progenitors (**new Fig. S11A**). MS identified 176 proteins with over a 2-fold increase in the ILF2-BioID2 group compared to the control group, suggesting their involvement in the ILF2 interactome (**new Table S6**). We excluded keratins from this list, as they are commonly regarded as contaminants in MS studies⁶. Gene enrichment analysis showed these proteins

were related to skin development, cell-cell junctions, and focal adhesion (**new Fig. S11B-E**). Furthermore, we overlapped these 176 proteins with those co-purified with SNHG26 in the RNA pull-down assay (Fig. 5A), identifying seven common proteins, two of which -ACTB and ANXA2 reside in the nucleus. ANXA2 has been shown to translocate into the nuclear in response to genotoxic agents and play important role in mitigating DNA damage¹¹. ACTB is part of chromatin-remodeling complexes and cooperates with different types of actin-binding proteins to regulate the transcriptional machinery¹². Our results suggest that these two proteins may be additional components of the ILF2/SNHG26 condensates in the nucleus. For convenience, we have included the results here and revised the result section on **page 16, line 305-316**.

Figure S11

Figure S11. Identification of proteins proximal to ILF2 by using BioID assay. (A) Western blot analysis of ILF2-BioID fusion protein in primary keratinocytes. **(B-E)** Functional enrichment analysis of proteins proximal to ILF2 in keratinocytes. **(F)** Overlap of the proteins identified by ILF2-BioID2 and SNHG26 pull down. **(G)** Subcellular location of the overlapped proteins in F.

Reviewer #2

The authors investigated the role of the lncRNA SNHG26 in inflammatory-to-proliferative state transition of keratinocyte progenitors. They identify this lncRNA as the relevant one by profiling lncRNA expression changes during human skin wound healing. Using different experimental models (human wound, KO mice, ASO-silencing, human keratinocytes, 3D wound model) and different technologies (bulk RNAseq, single cell RNA-seq, CIRP, RNA-pool down and MS, etc) they demonstrated that SNHG26 promote inflammatory-to-proliferative state transition of keratinocyte.

Mechanistically, they proposed that SNHG26 interacted with ILF2 and relocated it from inflammatory genomic loci, such as JUN, IL6, IL8, and CCL20, to the genomic locus of LAMB3, important for cytoskeleton keratinocytes remodeling.

The article is very interesting. The authors have performed many experiments to prove their hypothesis, and the conclusions are mostly supported by the experimental data shown. The results shown could be very useful for regenerative medicine.

My main concern is that most of the results presented are shown at RNA level, this is specifically related to the mechanistic part. To strength the data they should demonstrate that the beta 3 subunit of laminin, Jun, ILF2 and the target cytokines chemokines, are indeed modulated upon SNHG26KO or silencing at protein level in the wound or in the different models. The mechanisms identified would be further strengthened by combined FISH and immunofluorescent staining to visualize the special relationship between the lncRNA SNHG26 and the protein ILF2 in cells/tissue.

R.2: We are grateful for the constructive feedback and valuable suggestions provided by the reviewer. In response to these, we have made significant revisions to our manuscript and carried out additional experiments to assess the protein expression levels of Jun (**new Fig. 5K**), cytokines (**new Fig. 5P**), LAMB3 (**new Fig. 7J**), and ILF2 (**new Fig. S12B**). Moreover, we have conducted dual SNHG26 FISH (using RNAscope) and ILF2 immunofluorescent staining in human keratinocyte progenitors, visualizing a co-localization of SNHG26 RNA and ILF2 protein within the cell nucleus (**Fig. S10H, I**).

Additional points to clarify:

2.1 Fig1. SNHG26 expression is upregulated a NW7 (Fig1H) while its expression is barely detectable in normal skin. In human keratinocytes, however, seem to be expressed in normal conditions (fig1I and J). The authors should explain the difference of the expression level and give more detail about the keratinocyte progenitor conditions (proliferating/differentiating/confluency)

R.2.1: We maintain human primary keratinocytes at early passages and subconfluent (under 80% confluency) to prevent differentiation induced by confluency. In these cultures, over 65% of the cells are stem/progenitor cells, while the remaining 35% are in an early differentiation

state¹³. Compared to human homeostatic epidermis, cultured primary keratinocytes better reflect the cell states of wound healing and epidermal regeneration^{13, 14}. Consistent with this, we observed stronger SNHG26 expression in cultured keratinocyte progenitors, similar to the levels seen in wounds and higher than those in homeostatic human skin.

2.2 Fig2. The authors should better explain how they evaluated a 40% reduction in wound healing in SNHG26KO mice (fig2A). The healing rate is delayed, but why 40%. Wound healing of Ctrl-ASO mice behave differently by WT mice (fif2a and Fig2b). The authors should explain this discrepancy. Is the injected ASO in control suitable? Is there inflammation going on?

R.2.2: To investigate the healing rate in SNHG26-KO mice, we calculated the wound healing rate constant (day^{-1}) using a one-phase decay model in GraphPad, as described by Naik *et al*¹. The results showed that the healing rate constant was 0.48 ± 0.04 in WT mice compared to 0.29 ± 0.09 in KO mice, indicating a 40% delay in wound healing in SNHG26-KO mice.

As the reviewer noted, the wounds of wild-type mice treated with Ctrl-ASO (Fig. 2B) healed slower than those of untreated wild-type mice (Fig. 2A). While some variability between experiments is expected, we attribute this discrepancy primarily to the adverse effects of the intradermal injection of ASO and transfection reagent. Using a control group with non-targeting oligonucleotides and transfection reagent is essential to exclude non-specific effects unrelated to SNHG26 knockdown (KD). This allows us to evaluate the specific effects of SNHG26 KD, separate from the effects of ASO and transfection reagent injection.

2.3 Fig2. In fig2e the authors evaluated as markers for proliferation and differentiation Krt16 and Sprr1b, they also referred to references 21-23. However, these references do not refer at keratinocytes. I am not aware of a role of Sprr1b in keratinocytes migration. The authors should look at additional and specific markers for migration.

R.2.3: We apologize for the incorrect citation. Silencing of KRT16 have been shown to inhibit the proliferation of keratinocyte¹⁵. SPRR1B was shown to protect keratinocytes from excessive ROS by direct quenching via their cysteine residues and knockdown of SPRR1B significantly decreased the migration of keratinocytes¹⁶. We have updated the references to those reporting the roles of Krt16 and Sprr1b in the proliferation and migration of keratinocytes:

KRT16:

Chen, J.G., Fan, H.Y., Wang, T., Lin, L.Y. & Cai, T.G. Silencing KRT16 inhibits keratinocyte proliferation and VEGF secretion in psoriasis via inhibition of ERK signaling pathway. *Kaohsiung J Med Sci* 35, 284-296 (2019).

SPRR1B:

Vermeij, W.P. & Backendorf, C. Skin cornification proteins provide global link between ROS detoxification and cell migration during wound healing. *PLoS One* 5, e11957 (2010).

2.4 Fig5. The authors performed RNA pool down experiments. The authors should give more detail on the negative control they have used. In the method section they mention that they used the LacZ pool down. They should also show this experiment in fig 5b. Other bands of similar or higher intensity where pooled down (SNHG26-1), were these bands analyzed?

R.2.4: Thank you for pointing this out. The negative control shown in Fig. 5B (the lane on the far right) is the LacZ pull-down. We have revised the label of Fig. 5B to specify this. For this negative control, we inserted the LacZ cDNA sequence (Table S5), a bacterial gene, into the pcDNA3.1 vector and transcribed it *in vitro* using the MEGAscript™ T7 Transcription Kit. Proteins co-purified with LacZ RNA were considered non-specific bindings. In this study, we only analyzed the most distinct band between the SNHG26 and LacZ groups (the protein band at ~45 kDa) by mass spectrometry.

2.5 Fig5. In fig5e the authors should add an unrelated nuclear RNA as addition control for RIP?

R.2.5: Thank you for the suggestions. We have included MALAT1, a nuclear lncRNA, as an additional control. The results indicate that ILF2 does not interact with MALAT1 (**new Fig. 5E**).

Figure 5E

Figure 5E. ILF2 RIP followed by qRT-PCR analysis of copurified RNAs in human keratinocyte progenitors (n=3).

2.6 Fig5. Fig5f shows expression of ILF2 at mRNA level. Does ILF2 expression profile follows the one of SNHG26 during wound healing also at protein level? Is ILF2 expression specific for keratinocytes?

R.2.6: As suggested, we performed ILF2 immunofluorescent staining on human acute wounds. We observed that ILF2 is mainly expressed in epidermal keratinocytes (**new Fig. S12B, C**). Similar to its RNA expression dynamics (**Fig. 5F**), epidermal ILF2 protein expression is

upregulated in day-1 (NW1) and day-7 (NW7) wounds compared to intact skin, and returns to basal levels by day-30 (NW30) (**new Fig. S12B, C**), mirroring the trend of SNHG26 (**Fig. 1D**). While ILF2 expression is not exclusive to keratinocytes and is also present in dermal cells, its dermal expression does not change during the wound healing process. For convenience, we have included the results here:

Figure S12

Figure S12. (B) Immunofluorescence analysis of ILF2 in human acute wounds. Scale bar, 50 μ m. The white dashed line indicates the boundaries between epidermis and dermis. (C) Fluorescent intensity arbitrary units (AU) were analyzed by Image J ($n=3-6$).

2.7 The author also reported increased Jun transcription after siILF2 (fig5j), does Jun increase also at protein level?

R.2.7: We conducted immunoblot analysis on keratinocytes that had been transfected with ILF2 siRNA and subsequently stimulated with TNF α . We observed that the knockdown of ILF2 significantly reduced TNF α -induced JUN protein expression (**new Fig. 5K**), paralleling the changes seen in JUN mRNA levels (**Fig. 5J**) For reading convenience, we copied the results here:

Fig. 5K

Figure 5K. Western blot of JUN expression in human keratinocyte progenitors after ILF2 silencing followed by TNF α stimulation.

2.8 Fig5. The authors try to use different approached to demonstrate the link between SNHG26, Jun and ILF2. Most of the experiments are however carried on at transcriptional level, without showing the protein. For instance, fig5o they measure a significant bat small reduction of IL6 IL8 and CCL20 mRNA upon siLF2. Does this small reduction at mRNA level affect le level of these pro-inflammatory cytokines and chemokines?

R.2.8: Following the reviewer's suggestions, we performed ELISA assay to measure the protein levels of IL6, IL8 and CCL20 in the culture medium of keratinocytes transfected with siILF2 followed by TNFa treatment. We found that knockdown of ILF2 significantly decreased TNFa induced IL6, IL8 and CCL20 production by human keratinocyte progenitors (**new Fig. 5P**). For reading convenience, we copied the results here:

Figure 5P. ELISA analysis of IL6, IL8, and CCL20 proteins produced by human keratinocyte progenitors with ILF2 silencing followed by TNFa stimulation (n=3).

2.9 Fig6. The authors focused their attention on the LAMB3 locus. Is the expression level of LAMB3 decreased in SNHG26KO mice wound?

R.2.9: In previous version of this manuscript, we have shown the wound-edge epidermal expression of Lamb3 was significantly reduced in the Snhg26-KO mice compared to the wild-type mice at the RNA level. During the revision, we also conducted LAMB3 immunostaining on wounds of Snhg26-KO and WT mice. We observed that LAMB3 protein expression is primarily localized to the epidermis. Notably, there was a clear decrease in LAMB3 protein expression in the wound-edge keratinocytes of the of Snhg26-KO mice (**new Fig. 7J**). For reading convenience, we copied the results here:

Figure 7J

Figure 7J. Immunofluorescence staining of LAMB3 in the wound-edge tissues of WT mice and SNHG26-KO mice (n=9-11). The star marked the wound edge. The arrow heads indicate the LAMB3+ basal cells. Basal layer keratinocyte expression of LAMBS was quantified (right panel).

2.10 To better demonstrate the mechanism they have identified, would be interesting to visualize by combined FISH and immunofluorescent staining the special relationship between the lncRNA SNHG26 and the protein ILF2 in one or more experimental model they have used.

R.2.10: We have performed the dual staining of lncRNA SNHG26 and the ILF2 protein in human keratinocytes progenitors. The result revealed the co-localization of SNHG26 RNA and ILF2 proteins within a condensate-like structure in the cell nucleus (**Fig. S10H**).

Figure S10H. Dual RNAscope® fluorescence in situ hybridization and immunostaining to visualize the SNHG26 RNA (red) and ILF2 protein (green) in human keratinocyte progenitors. Nuclei were stained with DAPI. Scale bar, 5 μ m.

Reviewer #3

In this interesting report Li et al. study the function of the non-coding RNA SNHG26 during wound healing. The analysis of human skin samples during wound healing identify several conserved lncRNAs including SNHG26 predicted to be a positive regulator of proliferation and migration and a negative regulator of genes involved in inflammation. Knocking out the entire

SNHG26 genomic locus in mice they found wound healing impairment. The expression profile indicated that indeed SNHG26 is a positive regulator of migration and proliferation. The author also found an increase of macrophages during wound healing in the KO mice respect to controls. To understand this KO phenotype at cell type level they performed scRNA-seq. As this analysis pointed to Keratinocytes, the authors tested SNHG26 function with an ASO in human keratinocytes in vitro through transcriptome profiling and cell assays. With pull-down assay and RIP the authors showed the interaction between ILF2/NF45 and SNHG26. With Chip-seq they showed that ILF2/NF45 binding to the genome can be positively and negatively regulated by SNHG26. The authors also show the genomic chromatin binding of SNHG26 itself and identified that SNHG26 regulates the recruitment of ILF2 in the promoter of LAMB3, a well characterised genes involved in cell adhesion and migration. Unfortunately, there is a missing integration of the data between Chip-seq and ChIRP-seq to understand a broader functional connection between SNHG26 and ILF2.

Overall the manuscript contains very interesting results and a refreshing mix of omic data. However often the conclusions are not supported by the data.

Below several points that should be taken in account before publication:

R3: We are grateful for the reviewer's insightful comments, which prompted us to conduct a more comprehensive investigation during our revision process. In response, we performed extensive new experiments and analyses. These include analyzing marker gene expression to determine the healing phases of both human and murine acute wounds (**New Fig. S3A-E**) and evaluating the expression of SNHG26-neighboring genes, TOMM7 and FAM126A, to rule out the possibility that SNHG26 operates *in cis* (**New Fig. S10A-D**). Additionally, we performed CD68 immunostaining on murine wounds treated with SNHG26-ASO (**New Fig. S5E**) and conducted immunofluorescence staining to verify the presence of the four basal cell clusters identified by scRNA-seq in skin and wounds (**New Fig. S7C**).

To better understand the functional connection between SNHG26 and ILF2, we integrated ChIP-seq and ChIRP-seq results. As discussed in **R3.18**, the ILF2 ChIP-seq experiment shows that SNHG26 knockdown (KD) reduces ILF2 binding to growth-related gene loci. The SNHG26 ChIRP-seq revealed specific genomic binding sites of SNHG26. We identified LAMB3, PLEC, LINC02386, and DISP3 as common genes bound by SNHG26 and ILF2 (**New Fig. S14C**), with SNHG26 KD reducing ILF2 binding to these genes. ILF2 ChIP-seq results show that SNHG26 KD reduces ILF2 binding to the LAMB3 gene (**Fig. 5G**), which was confirmed by ILF2 ChIP-qPCR (**Fig. 6G**) and by ChIRP-seq demonstrating SNHG26's binding to the LAMB3 gene (**Fig. 6C**). These findings highlight the important role of LAMB3 in mediating the pro-proliferative and pro-migratory effects of SNHG26. Additionally, PLEC and DISP3, like LAMB3, have been reported to promote cell proliferation, suggesting they may also mediate SNHG26's pro-proliferative role through ILF2-induced expression^{17, 18}. Using a proximity-dependent biotin identification (BioID) assay followed by mass spectrometry analysis, we identified additional protein components, ACTB and ANXA2, in the SNHG26-ILF2 complex (**New Fig. S11**).

Furthermore, we found that SNHG26 not only redirects ILF2 genomic binding sites but also gains enhanced stability when bound to the ILF2 protein (**New Fig. S10J**). These new data have significantly solidified the proposed mechanism and support our conclusions. Please see our point-by-point response below.

3.1 "...The screening of LncRNAs identified a small nucleolar RNA host gene 26 (SNHG26)..."
This sounds like something is wrong for readers as small nucleolar RNA and LncRNA are quite different categories. The explanation of the complex locus should be done in the introduction.

R.3.1: We regret the previous oversight in our description. To clarify, the SNHG26 locus generates two distinct types of RNAs: the long non-coding RNA, referred to as Small Nucleolar RNA Host Gene 26 (SNHG26), and a small nucleolar RNA known as Snord93. We have updated the introduction section of our manuscript to accurately reflect this information (**page 4, lines 68-71**): *'Among these, small nucleolar RNA host gene 26 (SNHG26), a novel lncRNA derived from snoRNA host genes (SNHGs), emerged as a key regulator facilitating the transition from inflammation to proliferation in keratinocyte progenitors.'*

3.2 Relative to Fig1. The analysis of the human to mouse non coding RNAs conservation is generally tricky and a quick look of it for the human SNHG26 does not show an obvious result. This conservation analysis should be shown in supplementary data, including a sequence comparison of SNHG26.

R.3.2: Evolutionary conservation of lncRNAs have been characterized at multiple dimensions, including sequence, secondary structure, function, genomic position, and mechanism of action¹⁹. For SNHG26, we compared the sequences of human and mouse SNHG26 and noted poor sequence similarity (**new Fig. S2**). However, both the human and mouse SNHG26 genes are located between the TOMM7 and FAM126A coding genes, suggesting interspecies syntenic conservation (**Fig. 1C, Fig. S1A, B**). Additionally, our functional study demonstrates that both human and mouse SNHG26 play crucial roles in the proliferation, migration, and inflammatory response of keratinocytes (**Fig. S2-4**). Therefore, we propose that SNHG26 is a genomic position- and function-conserved lncRNA. For reading convenience, we copied the RNA sequence comparison results here:

3.3 Fig1.D NW1, NW7, NW30 in human and NW3, NW7, NW10 are assume to be the right time points to be defined as a inflammatory, proliferative and remodeling phases. However, the authors should show data in their settings to prove these associations.

R.3.3: We thank the reviewer for this excellent question. The wound healing process is typically characterized into three overlapping phases—inflammation, proliferation, and remodeling—based on the dynamics of various biological events. In our human wound healing model, we have characterized these biological events at each time point using gene expression profiles analyzed by bulk²⁰ and single-cell RNA-seq (unpublished data and **new Fig. S3A**). Our data show that inflammatory gene expression peaks at day-1 wounds (NW1), while cell mitosis-related gene expression peaks at day-7 wounds (NW7) (see Fig. 2f in Liu et al.²⁰). In the revised manuscript, we have included the expression dynamics of a panel of genes that reflect these biological processes: inflammation (S100A7 and S100A8), proliferation (MKI67), and remodeling (COL1A1 and COL3A1), peaking at NW1, NW7, and NW30, respectively (**new Fig. S3A**). This supports the association between the selected time points and the phases of wound healing. Similarly, in the mouse wound model, we observed peaked expression of inflammatory genes (Il6 and Il1a), a proliferative gene (Mki67), and a remodeling-related gene (Eln) at NW3, NW7, and NW10 wounds, respectively, linking these time points to the corresponding healing stages (**new Fig. S3B-D**). Overall, in this study, we utilized these sequential wound samples to demonstrate the transient upregulation of SNHG26 during wound healing. For convenience, we have included the results here:

Figure S3. (A) The expression of S100A7, S100A8, MKI67, COL1A1, and COL3A1 in single cell RNA sequencing data of human acute wounds. (B-E) qRT-PCR analysis of Il6, Il1a, Mki67 and Eln in murine acute wounds.

3.4 "...Notably, the SNHG26 gene locus also contains a small nucleolar RNA SNORD93 in both human and mice (Fig. 1C, Fig. S1A, B), but only SNHG26 showed increased expression in keratinocytes during wound repair (Fig. S1E, F)..." This is quite hard to understand. How is possible that the lncRNA (that contains the snoRNA) is differentially expressed while the contained snoRNA is not. Rt-pcr with one primer on the snoRNA and one in the remaining part of the lncRNA should help.

R.3.4: This is an interesting question. It has been shown that the levels of mature snoRNAs are rarely well correlated with the steady-state levels of their hosts, despite their shared transcription^{21, 22}. Various cellular mechanisms underpin the differential accumulation of the various transcripts produced from snoRNA host genes (SNHG), which has been recently reviewed²³. For example, SNHGs are preferential substrates of the nonsense mediated decay (NMD) pathway^{24, 25}; SNHGs often feature alternative transcription start site usage, which can tune snoRNA/host ratios²⁶. Moreover, the non-correlation between snoRNA and their host expression may be also due to the precise regulation of snoRNA biogenesis and maturation, which has been most studied in yeast, but unknown in mammals²³. Indeed, about ~21% of the SNHGs in the snoRNA database snoDB are labeled as lncRNAs, it was only until very recently that SNHGs gained interest beyond being not only simply hosts for snoRNAs, but also as carriers of independent functions²³, such as SNHG26 we have shown in this study. Therefore, understanding the regulation mechanisms of SNHG/snoRNA expression in the physiological relevant context would be an interesting direction for future study.

3.5 The strategy to generate the KO is not optimal. For instance it would be better to add a poli-A at the end of the first exon, to delete the expression of the lncRNA. Anyway, having removed the entire locus of the lncRNA might have impaired the regulation of surrounding genes if the lncRNA is working in "cis". At least this should be checked.

R.3.5: We appreciate the reviewer's attention to this important issue. In human keratinocyte progenitors, we show that SNHG26 knockdown did not affect the expression of the neighboring genes TOMM7 and FAM126A. Similarly, in mouse skin and wounds, Snhg26 KO did not change the expression of Tomm7 or Fam126a (**New Fig. S10A-D**). Thus, we conclude that SNHG26 is unlikely to act *in cis*, making the current strategy of generating KO mice suitable for understanding the function of SNHG26. For convenience, we have included the results here:

Figure S10. (A-B) qRT-PCR analysis of *Tomm7* and *Fam126a* mRNA expression in the skin and wound of WT and SNHG26-KO mice (n=5-8). (C-D) qRT-PCR analysis of *TOMM7* and *FAM126A* mRNA expression in the primary keratinocytes after SNHG26 knockdown (n=4).

3.6 How the ASO was used in vivo is not explained in the method. This is important as the author obtained a delivery specific to the epidermis and not the dermis (Figure S2F).

R.3.6: We have revised our methods section to provide more detailed information about the ASO treatment (page 2, lines 33-37, and see below). Briefly, we did not perform any epidermis-targeted delivery. The higher knockdown (KD) efficiency in epidermal cells may be due to the higher expression level of *Snhg26* and greater cell density in the epidermis compared to the dermis.

'To study the effect of SNHG26 knockdown on murine wound healing, SNHG26 Antisense LNA GapmeRs (SNHG26-ASO, 0.25 nmol per wound) or Control GapmeRs (Ctrl-ASO, 0.25 nmol per wound) were mixed with MaxSuppressor™ In Vivo RNA-LANCER II (Bioo Scientific) according to the manufacturer's instructions, in a total volume of 100 µl per wound. The mixture was injected into the space between the epidermis and dermis at four points around the wound edges in the mice immediately after wounding.'

3.7 Fig2c,d GSEA support the impact on the proliferation and migration. However it is not possible to claim anything based on GO term enrichment as the authors show "gene number" instead of adjpV (or pV at least). So far the author can only claim an effect of the two validated genes IL6, IL1b.

R.3.7: We agree with the reviewer and have revised Fig. 2C to show the p-values of the top five GO terms for the up- and down-regulated genes in *Snhg26* KO mouse wound-edge epidermis. For convenience, we have included the results here:

Figure 2C. Gene Ontology analysis of differentially expressed genes in the wound-edge epidermis of *Snhg26*-KO mice vs. WT mice.

3.8 The data to support the conclusions on exacerbated inflammatory response in KO are not enough. In Fig 2G there is a 2 fold enrichment of macrophages. However from the analysis provided on scRNA-seq is not possible to understand if these and other immune cells are increased in the wound time point in KO vs wt. the normalization on Fig3D is wrong as the percentage should be done on the total of cells and not on the total cell number in each single cluster.

R.3.8: We appreciate the reviewer's comment. Following the suggestion, we re-analyzed the changes in cell composition in WT and KO mice within our single-cell sequencing dataset and updated Fig. 3E accordingly. We found that the percentage of macrophages is higher in the wounds of KO mice compared to WT mice (**New Fig. 3D**), which is consistent with immunostaining results showing increased CD68+ cells in SHNG26-KO mice (**Fig. 2H**). Additionally, we observed a similar increasing trend for T cells (Th and gdT cells) and Langerhans cell (LC) in the KO mice wounds compared to WT mice wounds (**New Fig. 3D**). We also performed CD68 immunostaining on wounds treated with SNHG26-ASO or Ctrl-ASO. Consistent with the results from *Snhg26*-KO mice, the number of CD68+ macrophages significantly increased in wounds treated with SNHG26-ASO, suggesting that the enhanced immune response observed in SNHG26-KO directly results from SNHG26 depletion in epidermal cells (**New Fig. S5E**). Together, these data support the conclusion of an exacerbated inflammatory response in KO mice wound healing. For convenience, we have included the new results here:

Figure 3D. Comparison of skin and wound cell composition between the KO and WT mice. Red line indicates keratinocyte clusters with changed cell composition in the wound of KO mice. Blue line indicates immune cell clusters.

Figure S5E. Immunofluorescence analysis of CD68+ macrophages in mouse wounds injected with SNHG26-ASO (n=3). Scale bar, 20 μ m.

3.9 Fig3H,I The Data representation is confusing and, most importantly, does not allow for a statistical analysis. Related to Fig3g,h,i – if SNHG26 is expressed equally on the different Basal clusters why the analysis is not performed on the union of these cell clusters?

R.3.9: As suggested by the reviewer, we plotted the expression scores of proliferation, migration, and inflammation-related genes in the union of four basal keratinocyte clusters. This analysis clearly shows that migration and proliferation-related genes are significantly downregulated, whereas inflammation-related gene expression is significantly upregulated in the basal keratinocytes of Snhg26-KO mice wounds compared to WT mice wounds (**New Fig. 3H-J**). For convenience, we have included the new results here:

Figure 3.

Figure 3H-J. Violin plots showing keratinocyte migration (**H**), proliferation (**I**) and inflammation scores (**J**) in *Snhg26*-KO and WT mice wound basal keratinocytes. *** $P < 0.001$ by Mann-Whitney U test (**H-J**).

3.10 An intersection between the migration/proliferation genes from the microarray data from 1 and scRNA-seq would be informative. The same is true for inflammatory genes.

R.3.10: Following the reviewer's suggestion, we examined the expression of migration/proliferation genes (*Krt16* and *Spr1b*) and inflammatory genes (*Il6* and *Il1b*) in our single-cell RNA sequencing data, which were initially identified through microarray and qRT-PCR analysis of mouse wound-edge epidermis (**Fig. 2E, G**). Our scRNA-seq data confirmed the microarray and qRT-PCR results, showing a significant decrease in *Krt16* and *Spr1b* expression, and an increase in *Il6* and *Il1b* expression in the keratinocytes of KO mice wounds compared to WT mice wounds (**New Fig. S8E-H**).

Figure S8

Figure S8E-H. Violin plots showing the expression of *Krt16* (**E**), *Spr1b* (**F**), *Il6* (**G**), and *Il1b* (**H**) in the wound edge keratinocytes of *Snhg26*-KO and WT mice analyzed by scRNA-seq. *** $P < 0.001$ by Mann-Whitney U test (**E-H**).

3.11 From Method section - The genes for the migr/prolif score were taken from GO terms. Why inflammatory genes were not?

R.3.11: The genes for calculating the inflammation scores were from a summarized datasets from GO:0090594 and literature²⁷. We have clarified this in the revised method section (Page 8, line 145-151).

3.12 Related to Fig3g,h,i - From the analysis it seems that different basal cell populations do different thing during wound healing. The author should confirm this by histology the existence of this heterogeneity in skin.

R.3.12: As suggested, we performed immunofluorescence staining of the marker genes for these four basal keratinocyte clusters in the skin and wound edge of WT mice: KRT15 (Bas1), MKI67 (Bas2), GJB2 (Bas3) and KLF5 (Bas4), confirming the heterogeneity of basal layer keratinocytes (New Fig. S7C). Moreover, we also compared our results with previous reported epidermal basal cell subclusters in mice skin and wound by Xing Dai group²⁸. We found a high degree of similarity between these two studies: our Bas1 resembles Col17a1-Hi keratinocytes (both highly expressed Col17a1); Bas2 corresponds to the Proliferating basal cluster (both highly expressed MKI67); Bas3 aligns with the growth arrest (GA)-like cluster (both highly expressed Krt16); and Bas4 is similar to both GA-like (Both highly expressed Serpinb2, lfi202b, Ptgs2, lrf6) and early-response (ER)-like (Both highly expressed marker genes Atf3, Klf6) clusters.

For reading convenience, we copied the immunostaining results here:

Figure S7C. Immunofluorescence staining of the marker genes of basal keratinocyte clusters in the skin and wound edge of wild-type mice. HF: Hair follicle. Scale bar, 100 μ m. * wound edge.

3.13 Following this, I'm not sure if it is correct talk about "cell state transition" rather than something like to "activate cell population with different potencies".

R.3.13: We appreciate the reviewer's suggestion. We consider the four basal keratinocyte clusters (Bas1-4) as four basal cell states that may be interchangeable. For example, the Bas4 cluster, characterized by high expression of inflammatory genes, increases in wound of KO mice compared to WT mice (**Fig. 3D**). This rise in Bas4 cell numbers in KO wounds is unlikely due to the activation of the existing Bas4 population, as these cells do not proliferate to expand their numbers (**Fig. 3G, Fig. S7A**). Instead, it likely results from the transition of other basal cell clusters to Bas4 due to decreased anti-inflammation effect by SNHG26 in KO mice. Therefore, we believe it is more accurate to state that SNHG26 reprograms gene expression to facilitate 'cell state transition' rather than 'activating cell populations with different potencies,' although the distinction between these two statements is subtle.

3.14 "This ASO was designed to target the exons of SNHG26, and it led to a significant reduction in SNHG26 expression while leaving SNORD93 unaffected (Fig. S5A, B)" (row217), hard to understand how this is possible. Please explain.

R.3.14: As discussed in **R3.5**, the snoRNA and their host expression are not correlated, and it is unknown whether and how SNORD93 is processed from SNHG26. SNHG26 is not merely a precursor of SNORD93, but a lncRNA with independent expression and function, which is likely produced and maintained by mechanisms even independent of SNORD RNA biogenesis. Based on our results, the ASOs binds to SNHG26 RNA and directs RNase H-mediated cleavage of SNHG26, leaving SNORD93 unaffected, as it does not contain any binding sites for these ASOs.

3.15 Fig4j – it is har to understand how the quantification was performed as from the representative images it is not possible to see any difference.

R.3.15: Here we utilize the fluorescent vital dye CMFDA to label epithelial cells within wounds, allowing for longitudinal monitoring of wound closure and facilitating evaluation of re-epithelialization²⁹. In Fig. 4J, the white and red dashed lines indicate the initial wound edges at Day 0 and the newly formed epidermis front at Day 3, respectively. The re-epithelization of each wound was evaluated by the healing rate (%)=(Area within white circle- Area within red circle)*100/ Area within white circle. This analysis was performed for each wound on Day 3 and Day 5 post-injury. We can see that nuclear overexpression SNHG26 significantly promoted re-epithelization in human *ex vivo* wounds. We have clarified this analysis process

in the revised method section (page 12, line 233-235).

3.16 The results of MS relative to Fig5B should be put in a supplementary table. Is the band in the red rectangle only visible in SNHG26-1?

R.3.16: Thank you for this suggestion. We have added the MS results as a **new supplementary Table S5**. The total amount of proteins in SNHG26-2 sample was lower than in SNHG26-1 sample, so the band in the red rectangle is less discernible in the SNHG26-2 than the SHNG26-1 samples (Fig. 5B). However, with Western blot analysis that is much more sensitive to detect specific protein, we demonstrate that ILF2 protein is co-purified with SNHG26 in both replicates (Fig. 5C).

3.17 Fig5.D Are the differences between IgG and Anti-ILF2 for gapdh and 18s significant?

R.3.17: There is no significant (n.s.) difference for GAPDH and 18s between IgG and Anti-ILF2 groups. We have revised Fig. 5E to highlight this. For reading convenience, we copied the results here:

Figure 5E

Figure 5E. ILF2 RIP followed by qRT-PCR analysis of copurified RNAs in human keratinocyte progenitors (n=3).

3.18 Enrich R pathway analysis in FigS6G, H is important but not convincing. are there other GO term that are enriched? Are other GO terms enriched (including BP, CC, MF, Kegg terms)? MAPK signaling pathway is also a regulator of proliferation in keratinocytes. Interleukin2 pathway is in Ctlr-ASO and not in SNHG26.

R.3.18: We appreciate the reviewer's question, which prompted us to revisit the GO analysis

related to Fig. S6G, H. After performing GO:BP, GO:CC, GO:MF, KEGG, Reactome, Bioplanet enrichment analysis as suggested, we found that multiple cell growth-related pathways (e.g., EGFR1 pathway, DNA replication pre-initiation, G2/M transition, response to epidermal growth factor) were enriched in the Ctrl-ASO group. This indicates that ILF2 preferentially binds to cell growth-related gene loci in the presence of SNHG26 in keratinocytes (**new Fig. S12D**).

When SNHG26 was knocked down, ILF2 left these cell growth-related gene loci. However, its new binding sites did not exhibit a clear inflammation profile, as the reviewer pointed out. We did identify that MAPK signaling and cellular response to cytokine stimulus were enriched in these binding site-related genes (**new Fig. S12D**). One of the top candidates identified in the SNHG26-ASO group is JUN (**Fig. 5G, H**). Compared to other immediate early response genes to inflammatory stimuli, our ChIP-qPCR results show that ILF2 binds to the JUN gene, but not, or very minimally, to the IL6, IL8, and CCL20 genomic loci without inflammatory stimuli (TNF α treatment) (**Fig. 5I, O**). Therefore, we identified JUN but not these inflammatory gene loci in the ChIP-seq of keratinocytes with SNHG26 KD, likely due to the lack of inflammatory stimuli.

In summary, in growing keratinocytes without inflammatory stimuli, the ILF2 ChIP-seq experiment shows that SNHG26-KD reduces ILF2 binding to growth-related gene loci. The conclusion that SNHG26-KD enhances ILF2 binding to inflammatory genomic loci is better supported by ChIP-qPCR results in keratinocytes with both SNHG26 KD and TNF α treatment (**Fig. 5Q**). We revised the result part accordingly (*please see page 17, line 322-330*). For reading convenience, we copied the new analysis results here:

Figure S12. (D) Functional enrichment analysis of the genomic loci bound by ILF2 in Ctrl-ASO and SNHG26-ASO transfected human keratinocyte progenitors.

3.19 “SNHG26 hijacks ILF2 from promoter” in Fig5N does not make any sense.

R.3.19: We agree with the reviewer that this text is confusing and have removed it from Figure 5.

3.20 Fig5 contains interesting data however so far the conclusions proposed are not supported by the data. The data so far showed that SNHG26 interacts with ILF2, that the absence of

SNHG26 can prevent or favor the ILF2 binding. ILF2 binding on Jun, IL6, IL8 and CCL20 promoters is negatively regulated somehow by SNHG26.

R.3.20: We agree with the reviewer and have revised our conclusions to more accurately reflect our results. Please see the revised result section (page 14, line274) and the title of Fig. 5: 'Silencing SNHG26 increases the binding of ILF2 to inflammatory genomic loci'.

3.21 Are the loci selected in Fig.5P identified from the Chip-seq?

R.3.21: These loci were not identified in the ChIP-seq data. As discussed in R3.18, this may be because the ChIP-seq was performed on growing keratinocytes without inflammatory stimuli. Although SNHG26's inhibitory effects are absent, ILF2 binding to inflammatory genes seems to require additional signals activated by inflammatory stimuli.

3.22 Row 275 the use of the word "enriched" is wrong.

R.3.22: Thanks for the suggestion. We have deleted this word.

3.23 The authors failed to cite Wu TH, et PLoS One. 2019 PMID: 31022259. Where it was shown that ILF2/NF45 targets transcriptionally Jun.

R.3.23: We thank reviewer for this suggestion and have cited this work as the Reference 56 in current manuscript.

3.24 ChIRP-seq experiments seems suboptimal as only 43 genomic loci were found. This is an important issue as no intersection with the CHIP-seq in Fig 5 is shown (most likely as the intersection is null).

R.3.24: Our ChIRP-seq analysis was performed under a strict experiment condition (e.g., usage of high number of anti-sense DNA tiling probes divided into 'even' and 'odd' pools, nonspecific probes targeting LacZ RNA as controls etc³⁰). And the results are of high quality and specific, supported by a series of evidence: the 43 SNHG26 occupancy sites identified in this experiment exhibited a notable enrichment within genic regions; the preeminent SNHG26 RNA occupancy site was situated within its genic region (**see R3.26**); Multiple EM for Motif Elicitation (MEME) analysis revealed SNHG26's preference for occupying a GA-rich polypurine DNA motif (e-Value = 3.2×10^{-24} ; **Fig. 6E**), suggesting that SNHG26 accesses the genome through specific DNA sequences.

The low number of SNHG26 genomic binding loci are more likely due to its low abundance. Chromatin-associated lncRNAs are typically expressed at low copy numbers, ranging from 0.3 to 100 copies per cell³¹. A recent study demonstrated that the low expression of a lncRNA is crucial for its function, providing specificity to its target regulation³². We have estimated human skin and wound keratinocytes contain approximately 1.5-2 copies and 41-55 copies of SNHG26 transcripts per cell, respectively. The low copy number of SNHG26 is in line with its low genomic binding sites, reflecting its functional specificity.

Following reviewer's suggestion, we compared ILF2 ChIP-seq and SNHG26 ChIRP-seq results, to better understand the functional connection between SNHG26 and ILF2. We identified LAMB3, PLEC, LINC02386, and DISP3 as common genes bound by SNHG26 and ILF2 (**New Fig. S14C**). ILF2 ChIP-seq further show that SNHG26 KD reduces ILF2 binding to these genes. Interestingly, PLEC and DISP3, like LAMB3, have been reported to promote cell proliferation, suggesting they may also mediate SNHG26's pro-proliferative role through ILF2-induced expression^{17, 18}.

3.25 The hypothesis that SNHG26 facilitate or favor somehow the binding of ILF2 in different loci seems true. However it is hard to believe that SNHG26-dependent ILF2 localization on LAMB3 is an important mechanism rather than one of the few genes identified by ChIRP-seq regulating.

R.3.25: In the ChIRP-seq analysis, the LAMB3 genomic locus was identified as a top SNHG26 binding site with very high statistical confidence (p adjust value = 0.00046) (**Fig.6C, Table S8**). Additionally, ILF2 ChIP-seq revealed that the LAMB3 gene is also bound and regulated by ILF2, with SNHG26 knockdown resulting in reduced ILF2 binding to the LAMB3 gene, as confirmed by ILF2 ChIP-qPCR (**Fig. 5G, Fig. 6G**). We further validated the LAMB3 gene's critical role in keratinocyte migration, proliferation, and adhesion (**Fig.7A-E**). Consistent with this, we observed deficient LAMB3 expression in SNHG26 knockout mouse keratinocytes during wound repair (**Fig. 7H, I, New Fig. 7J**). Based on these findings, we consider LAMB3 to be a crucial mediator of the pro-proliferative and pro-migratory effects of the SNHG26-ILF2 axis.

3.26 Related to S6N - A general weak point of this manuscript is that the KO was obtained with the deletion of the entire locus of SNHG26. However it is clear that SNHG26 works also in "cis" on his own locus (the best peak in ChIRP-seq?). This type of regulatory regions can influence the expression of neighboring genes.

R.3.26: In ChIRP-seq, the preeminent SNHG26 RNA occupancy site situated within its genic region reflects the capture of SNHG26 that is being or newly transcribed but have not left its

genomic DNA locus (**Fig. 6C, Fig. S14A**). We have shown that SNHG26 does not act *in cis*, as knockdown of SNHG26 did not affect the expression of SNHG26-neighboring genes, TOMM7 and FAM126A in human keratinocyte progenitors (**Fig. S10C-D**); Snhg26 KO did not change Tomm7 or Fam126a expression either in mouse skin and wounds (**Fig. S10A-B**).

3.27 The statistics are present most of the times. However it is necessary to specify the number of replicates.

R.3.27: Thank you for the suggestions. We have included the number of replicates in each figure legend.

References:

1. Naik, S. *et al.* Inflammatory memory sensitizes skin epithelial stem cells to tissue damage. *Nature* **550**, 475-480 (2017).
2. Maldonado, H. *et al.* Systemically administered wound-homing peptide accelerates wound healing by modulating syndecan-4 function. *Nat Commun* **14**, 8069 (2023).
3. Eming, S.A., Martin, P. & Tomic-Canic, M. Wound repair and regeneration: mechanisms, signaling, and translation. *Sci Transl Med* **6**, 265sr266 (2014).
4. Tan, M.L.L., Chin, J.S., Madden, L. & Becker, D.L. Challenges faced in developing an ideal chronic wound model. *Expert Opin Drug Discov* **18**, 99-114 (2023).
5. Andergassen, D. & Rinn, J.L. From genotype to phenotype: genetics of mammalian long non-coding RNAs in vivo. *Nat Rev Genet* **23**, 229-243 (2022).
6. Hodge, K., Have, S.T., Hutton, L. & Lamond, A.I. Cleaning up the masses: exclusion lists to reduce contamination with HPLC-MS/MS. *J Proteomics* **88**, 92-103 (2013).
7. Wu, T.H., Shi, L., Lowe, A.W., Nicolls, M.R. & Kao, P.N. Inducible expression of immediate early genes is regulated through dynamic chromatin association by NF45/ILF2 and NF90/NF110/ILF3. *PLoS One* **14**, e0216042 (2019).
8. Zhang, X. *et al.* Interleukin enhancer-binding factor 2 promotes cell proliferation and DNA damage response in metastatic melanoma. *Clin Transl Med* **11**, e608 (2021).
9. Pandey, R.R. *et al.* Kcnq1ot1 antisense noncoding RNA mediates lineage-specific transcriptional silencing through chromatin-level regulation. *Mol Cell* **32**, 232-246 (2008).
10. Roux, K.J., Kim, D.I., Burke, B. & May, D.G. BioID: A Screen for Protein-Protein Interactions. *Curr Protoc Protein Sci* **91**, 19 23 11-19 23 15 (2018).
11. Madureira, P.A., Hill, R., Lee, P.W. & Waisman, D.M. Genotoxic agents promote the nuclear accumulation of annexin A2: role of annexin A2 in mitigating DNA damage. *PLoS One* **7**, e50591 (2012).
12. Visa, N. & Percipalle, P. Nuclear functions of actin. *Cold Spring Harb Perspect Biol* **2**, a000620 (2010).
13. Enzo, E. *et al.* Single-keratinocyte transcriptomic analyses identify different clonal types and proliferative potential mediated by FOXM1 in human epidermal stem cells. *Nat Commun* **12**, 2505 (2021).
14. Wang, S. *et al.* Single cell transcriptomics of human epidermis identifies basal stem cell transition states. *Nat Commun* **11**, 4239 (2020).
15. Chen, J.G., Fan, H.Y., Wang, T., Lin, L.Y. & Cai, T.G. Silencing KRT16 inhibits keratinocyte proliferation and VEGF secretion in psoriasis via inhibition of ERK signaling pathway. *Kaohsiung J Med Sci* **35**, 284-296 (2019).
16. Vermeij, W.P. & Backendorf, C. Skin cornification proteins provide global link between ROS detoxification and cell migration during wound healing. *PLoS One* **5**, e11957 (2010).
17. Zikova, M. *et al.* DISP3 promotes proliferation and delays differentiation of neural progenitor cells. *FEBS Lett* **588**, 4071-4077 (2014).
18. Perez, S.M., Brinton, L.T. & Kelly, K.A. Plectin in Cancer: From Biomarker to Therapeutic Target. *Cells* **10** (2021).
19. Guo, C.J. *et al.* Distinct Processing of lncRNAs Contributes to Non-conserved Functions in Stem Cells.

- Cell* **181**, 621-636 e622 (2020).
20. Liu, Z. *et al.* Integrative small and long RNA omics analysis of human healing and nonhealing wounds discovers cooperating microRNAs as therapeutic targets. *Elife* **11** (2022).
 21. Fafard-Couture, E., Jacques, P.E. & Scott, M.S. Motif conservation, stability, and host gene expression are the main drivers of snoRNA expression across vertebrates. *Genome Res* **33**, 525-540 (2023).
 22. Fafard-Couture, E., Bergeron, D., Couture, S., Abou-Elela, S. & Scott, M.S. Annotation of snoRNA abundance across human tissues reveals complex snoRNA-host gene relationships. *Genome Biol* **22**, 172 (2021).
 23. Monziani, A. & Ulitsky, I. Noncoding snoRNA host genes are a distinct subclass of long noncoding RNAs. *Trends Genet* **39**, 908-923 (2023).
 24. Colombo, M., Karousis, E.D., Bourquin, J., Bruggmann, R. & Muhlemann, O. Transcriptome-wide identification of NMD-targeted human mRNAs reveals extensive redundancy between SMG6- and SMG7-mediated degradation pathways. *RNA* **23**, 189-201 (2017).
 25. Lykke-Andersen, S. *et al.* Human nonsense-mediated RNA decay initiates widely by endonucleolysis and targets snoRNA host genes. *Genes Dev* **28**, 2498-2517 (2014).
 26. Nepal, C. *et al.* Dual-initiation promoters with intertwined canonical and TCT/TOP transcription start sites diversify transcript processing. *Nat Commun* **11**, 168 (2020).
 27. Jiang, Y. *et al.* Cytokinocytes: the diverse contribution of keratinocytes to immune responses in skin. *JCI Insight* **5** (2020).
 28. Haensel, D. *et al.* Defining Epidermal Basal Cell States during Skin Homeostasis and Wound Healing Using Single-Cell Transcriptomics. *Cell Rep* **30**, 3932-3947 e3936 (2020).
 29. Nasir, N.A.M., Paus, R. & Ansell, D.M. Fluorescent cell tracer dye permits real-time assessment of re-epithelialization in a serum-free ex vivo human skin wound assay. *Wound Repair Regen* **27**, 126-133 (2019).
 30. Chu, C., Quinn, J. & Chang, H.Y. Chromatin isolation by RNA purification (ChIRP). *J Vis Exp* (2012).
 31. Wu, M., Yang, L.Z. & Chen, L.L. Long noncoding RNA and protein abundance in lncRNPs. *RNA* **27**, 1427-1440 (2021).
 32. Jachowicz, J.W. *et al.* Xist spatially amplifies SHARP/SPEN recruitment to balance chromosome-wide silencing and specificity to the X chromosome. *Nat Struct Mol Biol* **29**, 239-249 (2022).

REVIEWERS' COMMENTS

Reviewer #1 (Remarks to the Author):

I appreciate the authors' responses. The authors have added sufficient details.

Reviewer #2 (Remarks to the Author):

The manuscript is highly interesting and effectively demonstrates the role of lncRNA SNHG26 in wound healing. The authors identify the molecular network involved and show that the mechanism is conserved between humans and mice. This is a crucial aspect, paving the way for new studies in the field. The authors have addressed most of the points raised, clarified numerous aspects, and included new experiments. In my opinion, the conclusions are now stronger and better supported by the data presented.

Reviewer #3 (Remarks to the Author):

The authors substantially strengthened the manuscript with new experiments to answer to all reviewers' comments. The added results strongly help to support the authors' conclusions. This manuscript describes, through a precise molecular analysis, the functional role of SNHG26 lncRNA in transition from an inflammatory phase to a proliferative phase, an important cellular process in the field of tissue repair. For these reasons I strongly support the publication of this work.